# Viral coinfection promotes tuberculosis immunopathogenesis by type I IFN signaling-dependent impediment of Th1 cell pulmonary influx

Tae Gun Kang [1,2,7], Kee Woong Kwon [3,7], Kyungsoo Kim[1,4], Insuk Lee [5], Myeong Joon Kim[1,2], Sang-Jun Ha [1,2✉] & Sung Jae Shin [3,6✉]

Tuberculosis (TB), caused by *Mycobacterium tuberculosis* (Mtb), is often exacerbated upon coinfection, but the underlying immunological mechanisms remain unclear. Here, to elucidate these mechanisms, we use an Mtb and lymphocytic choriomeningitis virus coinfection model. Viral coinfection significantly suppresses Mtb-specific IFN-γ production, with elevated bacterial loads and hyperinflammation in the lungs. Type I IFN signaling blockade rescues the Mtb-specific IFN-γ response and ameliorates lung immunopathology. Single-cell sequencing, tissue immunofluorescence staining, and adoptive transfer experiments indicate that viral infection-induced type I IFN signaling could inhibit CXCL9/10 production in myeloid cells, ultimately impairing pulmonary migration of Mtb-specific CD4$^+$ T cells. Thus, our study suggests that augmented and sustained type I IFNs by virus coinfection prior to the pulmonary localization of Mtb-specific Th1 cells exacerbates TB immunopathogenesis by impeding the Mtb-specific Th1 cell influx. Our study highlights a negative function of viral coinfection-induced type I IFN responses in delaying Mtb-specific Th1 responses in the lung.

[1] Department of Biochemistry, College of Life Science & Biotechnology, Yonsei University, Seoul 03722, Republic of Korea. [2] Brain Korea 21 (BK21) FOUR Program, Yonsei Education & Research Center for Biosystems, Yonsei University, Seoul 03722, Republic of Korea. [3] Department of Microbiology, Graduate School of Medical Science, Brain Korea 21 Project, Yonsei University College of Medicine, Seoul 03722, Republic of Korea. [4] Institute for Breast Cancer Precision Medicine, Yonsei University College of Medicine, Seoul, Republic of Korea. [5] Department of Biotechnology, College of Life Science & Biotechnology, Yonsei University, Seoul 03722, Republic of Korea. [6] Institute for Immunology and Immunological Disease, Yonsei University College of Medicine, Seoul 03722, Republic of Korea. [7] These authors contributed equally: Tae Gun Kang, Kee Woong Kwon. ✉email: sjha@yonsei.ac.kr; sjshin@yuhs.ac

Tuberculosis (TB) remains a notorious infectious disease caused by a single bacterial agent (*Mycobacterium tuberculosis*, Mtb). It has been estimated that almost one-fourth of the world's population is latently infected with Mtb and that nearly 10% of these individuals will progress to an active state[1,2] during their lifetime.

Concurrent or sequential infections with other pathogens may alter TB progression toward exacerbated pulmonary pathology, and additional infections in patients with active TB disease also hinder efforts to reduce the worldwide Mtb prevalence. For the best example, coinfection with human immunodeficiency virus (HIV) and Mtb causes detrimental outcomes with high mortality among active TB patients and increases the possibility of reactivation from latent TB infection[3–5]. Additionally, influenza and helminth infections also exacerbate TB pulmonary pathology[6,7]. One possible reason for this worsened Mtb pathogenesis is that helminth coinfection directs a T helper 2 (Th2) cell-skewed immune response and disrupts the essential protective T helper 1 (Th1) cell response[8,9]. Moreover, influenza infection followed by bacterial infection due to several bacteria has been found to be a major reason for increasing mortality worldwide[10,11]. In animal models, influenza A virus infection has been shown to cause a substantial increase in susceptibility to secondary bacterial infection, including Mtb infection, resulting in increased bacterial loads and tissue damage and decreased survival[12–14]. However, the underlying mechanism of the alteration of the immune response to TB upon coinfection remains insufficiently understood due to limited clinical and preclinical experimental data.

Type I interferons (IFNs, mainly IFN-α and IFN-β) have recently gained increasing attention regarding their function of TB pathogenesis. While type I IFNs are essential for protection against virus infection[15,16], they were recently found to cause immunopathological reactions in Mtb infection[17]. Data from both mouse experimental models and humans support the detrimental function of type I IFNs in TB outcomes. The overexpression of type I IFN-related genes was detected in blood at early stages of Mtb infection prior to progression to active disease, suggesting that systemic activation of the type I IFN response is the critical determinant for the onset of active disease from the latent state[18,19]. Additionally, transcriptional profiles of blood samples from patients with active TB yielded an upregulated type I IFN-inducible gene signature, which correlated with disease progression and curtailed Mtb treatment success[20]. These findings were also validated in diverse patient cohorts with different genetic and geographical backgrounds[21–23]. In mice infected first with influenza virus, a major in vivo inducer of type I IFN production, and then Mtb infection, the pulmonary pathology deteriorated in a type I IFN-dependent manner[24]. Moreover, type I IFNs derived from macrophages have been shown to contribute to Mtb-induced macrophage cell death, and blockade of type I IFN signaling has been shown to enhance the activity of rifampin[25]. Furthermore, the magnitude of Mtb-induced type I IFN production in mice has been reported to differ according to the Mtb strain. Mice infected with clinically isolated Mtb strains such as HN878 and BTB02-171 displayed increased type I IFN production compared to that of mice infected with laboratory-adapted strains such as H37Rv[26–28]. The underlying mechanism of differential type I IFN production levels for different Mtb strains in an in vitro macrophage infection system was associated with their capacity to induce mitochondrial stress, reactive oxygen species generation and host mitochondrial DNA release into the cytosol[29]. Likewise, diverse molecular functions of type I IFN in Mtb pathogenesis have been proposed based on the altered innate function of macrophages and neutrophils[30]. However, the effect of type I IFNs on the pulmonary immune system, in particular T cell immunity, is still unclear in the pathogenesis of Mtb coinfection with other pathogens.

Here, we investigate the underlying mechanisms by which viral infection-induced type I IFN signaling disrupts the Mtb-specific Th1 response prior to recruiting IFN-γ-producing T cells to the lungs from draining lymph nodes (dLNs). Mice were initially infected with clinically isolated Mtb K and then coinfected with *lymphocytic choriomeningitis virus* (LCMV). Coinfected mice exhibited significant increases in bacterial loads and hyperinflammation in the lungs with a dramatic decrease in the Mtb-specific Th1 cell response compared to those of mice infected with Mtb alone. These detrimental effects of viral coinfection were abrogated by the blockade or absence of type I IFN signaling. Of note, single-cell RNA sequencing (scRNA-seq) and lung tissue immunostaining showed that the expression of CXCL9/10, chemokines that recruit effector T cells from dLNs, in a pulmonary macrophage subset was significantly repressed in coinfected mice and that the expression of these chemokines was restored when the type I IFN receptor was blocked. Surprisingly, the frequencies of Mtb-specific CD4+ T cells in dLNs were not affected even after virus coinfection, suggesting that virus-induced type I IFN signaling repressed the expression of pulmonary CXCL9/10, leading to subsequent failure of Mtb-specific Th1 cells to migrate from dLNs to infected lung tissues. Indeed, a predominance of type I IFN signaling abrogated IFN-γ-mediated CXCL9 expression. Overall, our study demonstrates that viral infection during Mtb progression promotes pulmonary immunopathology along with uncontrolled bacterial growth, suggesting an underlying mechanism by which virus-induced type I IFNs inhibit chemokine-dependent pulmonary migration of Mtb-specific IFN-γ-producing T cells.

## Results

**Coinfection with LCMV exacerbates the pathology of mice infected with Mtb.** We first established a mouse coinfection model to investigate the effect of virus infection on TB progression in vivo. Mice were aerogenically infected with Mtb K first and later coinfected with LCMV Armstrong (Arm), inducing acute infection, or LCMV Clone 13 (CL13), causing chronic infection, at 14 days post Mtb infection. Mice were sacrificed at four weeks post Mtb K infection (Fig. 1a). Large inflammatory gross pathology and H&E-stained sections of granulomas with remarkable tissue destruction and central necrosis characteristics were observed in coinfected mice compared to those in mice infected with Mtb K alone (Fig. 1b, c, and Supplementary Fig. 1a and b) regardless of the virus strain. Additionally, abundant acid-fast bacillus (AFB)-stained bacteria were detected in these necrotic granuloma lesions (Fig. 1d and Supplementary Fig. 1c). In line with exacerbated lung pathology, the weight loss of coinfected mice was greater than that of Mtb K-only infected mice (Fig. 1e). Also, larger inflammatory area was detected in the lungs of coinfected mice (Fig. 1f and Supplementary Fig. 1d). These results are also supported by elevated bacterial loads in other organs of coinfected mice, such as the spleen and liver, as well as in the lung (Fig. 1g and Supplementary Fig. 1e).

Due to the earlier occurrence of death with LCMV CL13 coinfection (Supplementary Fig. 1f), LCMV Arm was used in subsequent experiments as the coinfection pathogen model. We further analyzed the dose-dependent effect of bacteria or viruses on pulmonary pathology. When mice were infected with a lower dose of Mtb or LCMV Arm (Supplementary Fig. 1g–i), they also exhibited a similar range of pulmonary pathology. In summary, viral coinfection during Mtb infection exacerbated pulmonary pathology and increased bacterial loads in vivo.

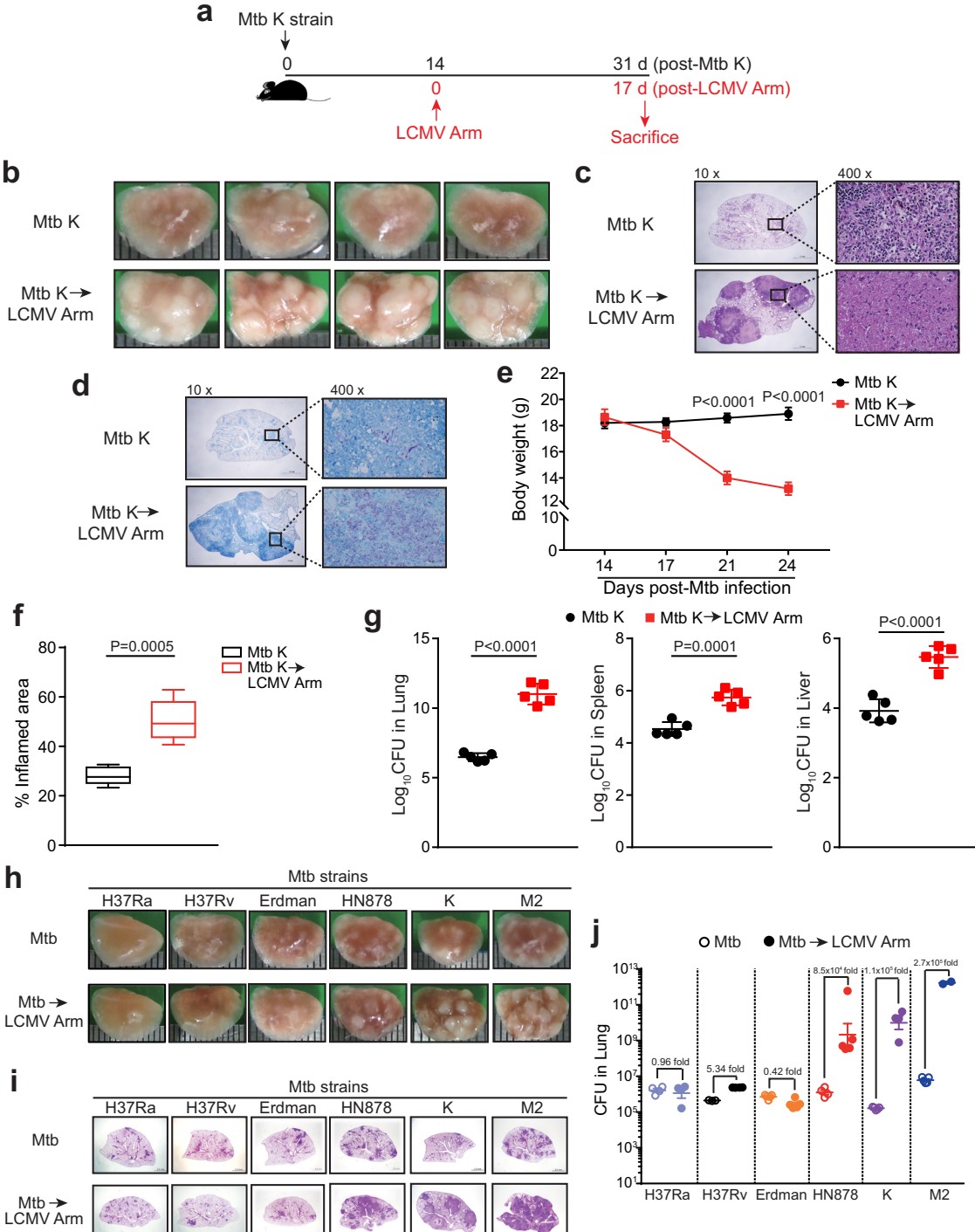

**Fig. 1 Severe immunopathology with necrotic granulomas in Mtb K and LCMV Arm-coinfected mice and differential pathogenesis of coinfected mice depending on the Mtb strain. a** C57BL/6 mice were infected with Mtb alone or with Mtb followed by subsequent infection with LCMV Arm at 14 days post Mtb infection. At 31 days post Mtb infection, the mice were sacrificed. **b** Gross pathology or **c**, H&E staining of lungs in each group. **d** Acid-fast staining of the lung in each group. (10x, scale bars, 2 mm; 400x, scale bars, 50 μm). **e** The weight of mice at the indicated time points ($n = 4$ or 5). **f** The data represent the percentage of the superior lobe of the right lung showing inflammation and are shown as center, bounds of box-and-whisker and percentile plots showing the minimum and maximum values ($n = 5$). **g**, Bacterial load in the organs of each group ($n = 5$). C57BL/6 mice were infected with the indicated Mtb strains. Some mice from each group were subsequently infected 2 weeks later with LCMV Arm. At 4.5 weeks post Mtb infection, the mice were sacrificed. **h** Gross pathology or **i** H&E staining of lungs in each group. Scale bars, 2 mm. **j** Bacterial load in the lungs of each group. Data were analyzed by **e–g**, two-tailed unpaired Student's *t* test. Graphs show the mean ± SEM. **b**, **c**, **d**, **f**, **g**, The data are representative of at least two independent experiments or **h–j**, a single experiment. Source data are provided as a Source Data file.

Next, we infected mice with diverse Mtb strains (H37Ra, H37Rv, Erdman, HN878, K, and M2) to investigate a possible association between aggravated immunopathology and different Mtb strains. Mice infected with HN878, K, or M2 showed exacerbated pathology along with excessive bacterial growth in the lung, whereas those infected with H37Ra, H37Rv, or Erdman showed less severe pathological profiles (Fig. 1h–j) and bacterial loads comparable to those with Mtb infection alone. Thus, pathology exacerbation was dependent on the Mtb strain.

**Neutrophilia and a diminished Th1 response are distinct features during coinfection.** We next investigated the immunological factors related to severe pathology in a coinfection model. In line with recent findings regarding the correlation between neutrophil infiltration and exacerbated pulmonary pathology[31,32] and our results above, we found a striking and gradual increase in the frequency and number of neutrophils at 7 days post LCMV Arm infection (Fig. 2a), while no significant differences in monocyte populations were observed (Supplementary Fig. 2b) between coinfection and Mtb infection alone. We also investigated other immune cell populations that infiltrated Mtb-infected lungs following viral coinfection. The frequencies of CD4+ and CD8+ T cells and dendritic cells (DCs) in coinfected mice dramatically decreased, while the frequencies of NKG2D+ cells among CD8+ T cells increased at 7 days post LCMV Arm infection (Supplementary Fig. 2b).

Levels of proinflammatory cytokines in Mtb-infected lung lysates, including IL-6 and TNF, were significantly upregulated (Fig. 2b), while IL-10 levels in the coinfected group were comparable with those in the Mtb-infected group at 7 days post LCMV Arm infection (Supplementary Fig. 2c). No significant differences in the levels of other cytokines, including IL-1α and IL-1β, were detected between the coinfected and Mtb-infected groups (Supplementary Fig. 2d). Previous studies have reported a detrimental effect of type I IFN during Mtb infection due to increased production of CCL2, which is mainly responsible for neutrophil infiltration to the infection site[33,34]. In line with these results, IFN-α, CCL2, and CXCL1 levels in the lungs of the coinfected group were higher and more sustained than those in the lungs of the Mtb-only infected group (Fig. 2c, d). Taken together, these results suggest that coinfection leads to excessive inflammation, including elevated neutrophil populations, and sustained high levels of chemokines and IFN-α.

The protective function of Mtb-specific IFN-γ-producing T cells for early host resistance to Mtb is well documented. We analyzed IFN-γ levels in the lungs of mice at 7 days post LCMV Arm infection. The overall IFN-γ levels in lung lysates decreased more in the coinfected group than in the Mtb-only infected group (Fig. 2e). Next, we determined the frequency of D$^b$Mtb32$_{92\text{-}102}$-specific CD8+ T cells by tetramer staining. These cells were barely detected in the coinfected group, in contrast to the Mtb-only infected group (Fig. 2f). Ex vivo restimulation with ESAT-6 or Mtb32 pools containing all peptides for functionality analysis of Mtb-specific T cells from among suspended single lung cells implied that ESAT6-specific CD4+ T cells from the Mtb-infected group produced IFN-γ and TNF at abundant levels, whereas the function of ESAT6-specific CD4+ T cells in coinfected mice dramatically decreased at 7 days post LCMV Arm infection. Additionally, the frequency of IFN-γ- and TNF-producing Mtb32-specific CD8+ T cells decreased (Fig. 2g, h). Hence, not only the number of T cells but also the function of Mtb-specific T cells dramatically decreased in the coinfected group compared to those in the Mtb-infected group. Notably, even without ex vivo restimulation, IFN-γ expression was still observed in CD4+ T cells of the Mtb-infected group but not in those of the

coinfected group (5.3% vs. 0.3%), indicating the presence of an Mtb-specific Th1 response in the lung during infection with Mtb alone. In addition, we further analyzed the kinetics of IFN-γ and ESAT6-specific T cell responses. At 14 days post LCMV Arm infection, both total IFN-γ and ESAT6-specific T cell responses in the lung were similar between the Mtb K alone-infected and coinfected groups (Supplementary Fig. 2e, f), suggesting that ESAT6-specific T cell responses were transiently delayed at 7 days post LCMV Arm infection. Taken together, these data suggest that the absence of Mtb-specific IFN-γ-producing T cells and the increased neutrophil levels contribute to the severity of TB outcomes in Mtb–virus coinfected mice.

**Preexisting Mtb-specific Th1 cells in the lung restrain virus coinfection-induced exacerbation of TB pathogenesis.** Next, using BCG vaccination and time-dependent viral infection models, we examined whether diminished Th1 responses are a major cause of viral coinfection-driven TB immunopathology or a consequent outcome. The BCG vaccine, the only available TB vaccine, elicits T cell responses in vivo[35] by early augmentation of Mtb-specific IFN-γ-producing T cell recruitment to the lung. Vaccination with BCG Pasteur 1173P2 was performed three months before Mtb infection (Supplementary Fig. 3a). At 14 days post Mtb infection, Mtb-infected mice without BCG vaccination barely elicited IFN-γ responses in the lung, whereas Mtb-infected mice with BCG vaccination showed significantly increased frequencies and numbers of Mtb-specific CD4+ T cells that were able to produce IFN-γ upon restimulation with purified protein derivative (PPD) (Supplementary Fig. 3a). Despite viral coinfection, the BCG-vaccinated group displayed less severe pulmonary pathology, fewer granuloma lesions (Supplementary Fig. 3b and 3c), lower bacterial loads in the lung (Supplementary Fig. 3d), and less neutrophil infiltration (Supplementary Fig. 3e) than the unvaccinated group. Additionally, cells from the BCG-vaccinated and coinfected group secreted significantly more IFN-γ (Supplementary Fig. 3f, g) upon restimulation with PPD at 21 days post Mtb infection than cells from the unvaccinated and coinfected groups, suggesting that the BCG-primed generation of Mtb-specific T cells in the lung might ameliorate the exacerbated pulmonary pathology induced by viral coinfection. In the same context, we further investigated whether the exacerbated pathology and elevated Mtb loads were also influenced by preexisting IFN-γ-producing T cells at different time points of LCMV infection (Supplementary Fig. 3h). At 21 days Mtb post-infection, when Mtb-specific IFN-γ-producing T cells already existed in the lung (Supplementary Fig. 3i), LCMV Arm coinfection had no detrimental effect on TB pathogenesis, while only mice infected with LCMV Arm at 14 days after Mtb infection showed exacerbated gross pathology (Supplementary Fig. 3j) and more severe granuloma lesions in the lung, with necrosis characteristics and a high bacterial burden (Supplementary Fig. 3k, l). Overall, LCMV Arm coinfection-induced TB immunopathogenesis occurs in a time-specific manner, indicating that viral coinfection-driven TB immunopathology is a result of significantly hindered Th1 responses at a specific time point.

**Type I IFN signaling blockade rescues necrotic immunopathology and Mtb-specific T cell responses in coinfected mice.** Next, since a positive correlation between the type I IFN signature and TB immunopathology in TB-susceptible mouse models[36–38] have been reported and we observed elevated and sustained IFN-α levels in lung lysates (Fig. 2c), we investigated the effect of type I IFN induced by LCMV Arm on TB immunopathology. Therefore, we blocked in vivo type I IFN receptor (IFNAR-1) signaling at specific time points (Fig. 3a) to investigate the relationship between type I IFN signaling and the pulmonary pathology induced by

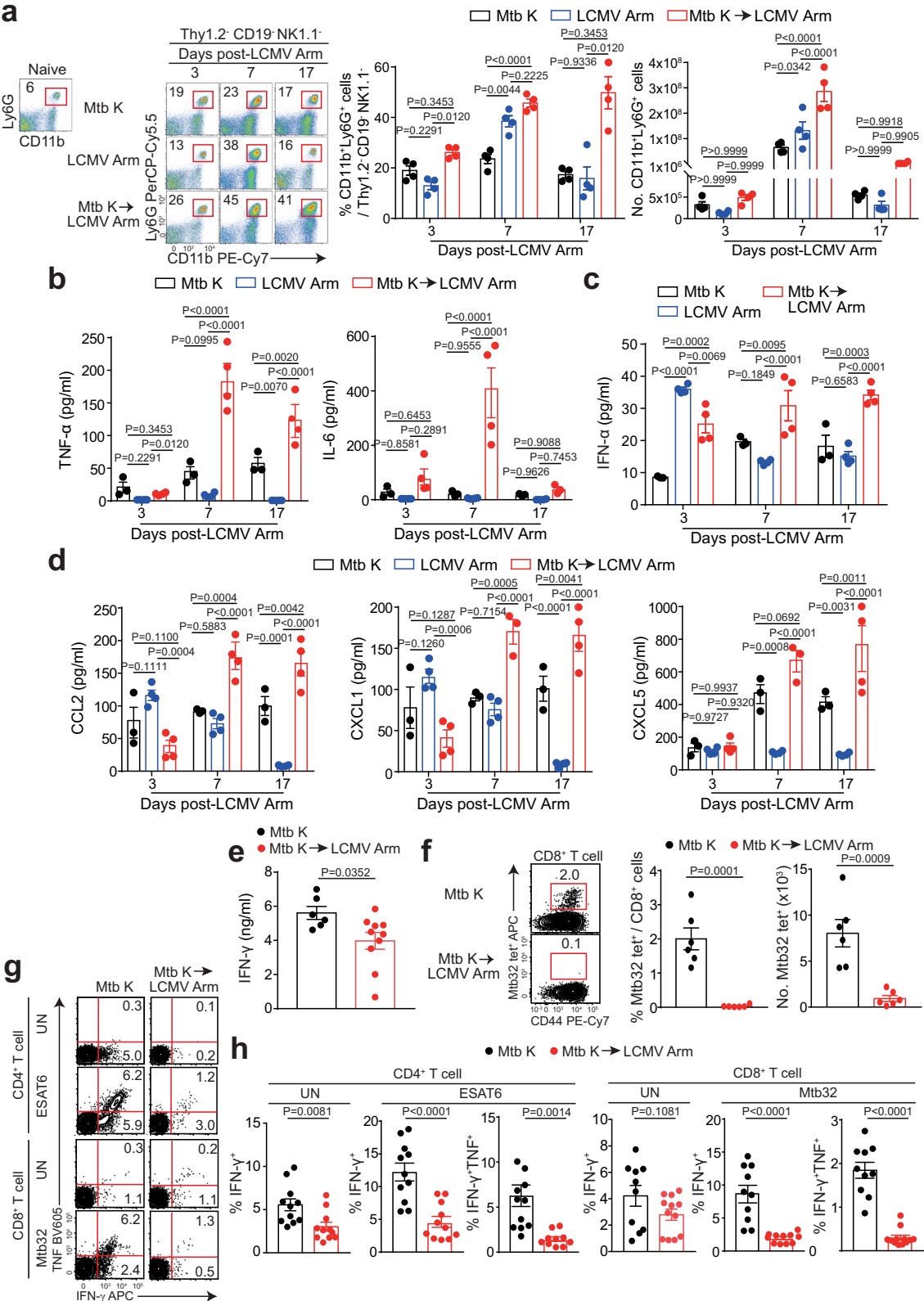

LCMV Arm infection. The coinfected group with IFNAR-1 blockade showed significantly reduced pulmonary pathology exacerbation comparable to the pathology in the control group (Fig. 3b, c, and Supplementary Fig. 4a). The lungs of mice in this group also had markedly reduced bacterial loads (Fig. 3d). Furthermore, coinfected mice with curtailed Mtb-specific Th1 immunity regained CD4$^+$ and CD8$^+$ T cell functionality both without and with ex vivo restimulation, according to both their frequency and absolute number (Fig. 3e, f). Consistent with the above results, the pulmonary pathology (Fig. 3g, h) and lung bacterial burden of type I IFN receptor knockout (*Ifnar1*$^{-/-}$) mice were remarkably reduced (Fig. 3i), indicating that type I IFN signaling induced by LCMV Arm infection is a major determinant of TB immunopathology upon viral coinfection.

**Fig. 2 Neutrophilia and increased inflammatory cytokine levels along with significantly reduced Mtb-specific T cell responses in coinfected mice.**
C57BL/6 mice were infected with Mtb. After 14 days, some mice were subsequently infected with LCMV Arm. **a** Frequency and number of $CD11b^+Ly6G^+$ cells. The numbers in the plot indicate the percentage of $CD11b^+Ly6G^+$ cells ($n = 4$). The protein levels of **b** TNF and IL-6 and **c** IFN-α in lung homogenates were analyzed by CBA and ELISA, respectively ($n = 3$ or 4). **d** Chemokine levels in lung homogenates were analyzed by multiplex analysis ($n = 3$ or 4). **e** At 7 days after LCMV Arm infection, the IFN-γ level in lung homogenates was analyzed by ELISA ($n = 6$ or 10). **f** Frequencies of $D^bMtb32_{93-102}$-specific $CD8^+$ T cells identified by tetramer staining were analyzed by flow cytometry. The numbers in the plots indicate the percentage of tetramer-positive cells among $CD8^+$ T cells ($n = 6$). **g–h**, Lung lymphocytes were isolated at 21 days post Mtb infection and analyzed by flow cytometry. The frequencies of IFN-γ- and TNF-producing $CD4^+$ T cells or $CD8^+$ T cells were analyzed by flow cytometry. Data were analyzed by two-tailed unpaired Student's $t$-test. Plots show the mean ± SEM ($n = 10, 11, 12$). **a–d** These data are representative of at least two experiments or **e–h** pooled from two or three independent experiments. **a–d** Data were analyzed by two-way ANOVA with *post hoc* Tukey's test or **e–h**, two-tailed unpaired Student's $t$-test. Graphs show the mean ± SEM. Source data are provided as a Source Data file.

We also detected elevated levels of several factors in coinfected mice. Neutrophils are the most abundant cells during TB and have been shown to contribute to pulmonary pathology exacerbation. $CD8^+$ T cells have also been shown to trigger immunopathology through a NKG2D-dependent mechanism in a mouse coinfection model with LCMV and *Leishmania major*[39]. Here, the levels of $CD8^+$ and $NKG2D^+$ $CD8^+$ T cells also increased in coinfected mice (Supplementary Fig. 2b). We performed in vivo blockade or depletion of the above factors at early time points to investigate the effects of neutrophils, immunosuppressive IL-10, and $NKG2D^+$ $CD8^+$ T cells on pulmonary pathology (Supplementary Fig. 5a). In contrast to blocking type I IFN signaling, the regulation of $Gr-1^+$ cells, including neutrophils, IL-10 signaling and $CD8^+NKG2D^+$ cells, did not affect the pulmonary pathology (Supplementary Fig. 5b, c) or lung bacterial burden of coinfected mice (Supplementary Fig. 5d). Furthermore, since the levels of inflammatory cytokines, such as IL-6 and TNF, and neutrophils increased at 7 days post LCMV Arm coinfection (Fig. 3 and Supplementary Fig. 2b), we also analyzed the effect of depletion or blockade of each factor (Supplementary Fig. 5e). The pulmonary pathology and lung bacterial burden were not diminished by blockade or depletion, and even worsened immunopathology was observed compared to that in isotype control group (Supplementary Fig. 5f–h). Surprisingly, unlike pre-blocking type I IFN signaling before viral coinfection (Fig. 3), the blockade of type I IFN signaling after LCMV Arm coinfection had no effect on TB immunopathology (Supplementary Fig. 5g), indicating that TB-associated immunopathology occurred upon viral coinfection once type I IFN signaling was activated.

Although $CD11b^+Ly6G^+$ cells are generally considered neutrophils, these cells have also been shown to suppress the adaptive immune response in specific environments, such as the tumor microenvironment. These cell populations are therefore defined as myeloid-derived suppressor cells (MDSCs)[40]. $CD11b^+Ly6G^+$ cells have also been shown to have a suppressive function in Mtb infection[41,42]. To investigate their characteristics in our mouse TB model, we isolated $CD11b^+Ly6G^+$ cells from the lungs of Mtb-infected and coinfected mice and measured the bacterial burden in these cells (Supplementary Fig. 5i). The $CD11b^+Ly6G^+$ cells of coinfected mice showed higher bacterial loads than those of Mtb-infected mice (Supplementary Fig. 5j). We also cocultured naïve T cells with isolated $CD11b^+Ly6G^+$ cells in the absence or presence of CD3 and CD28 stimulation to investigate the suppressive capacity of these $CD11b^+Ly6G^+$ cells. T cells cultured with only CD3 and CD28 stimulation showed a proliferation index similar to that of those cultured with $CD11b^+Ly6G^+$ cells isolated from infected lungs (Supplementary Fig. 5k, l). In summary, here, $CD11b^+Ly6G^+$ cells did not contribute to the pulmonary pathology exacerbation and acted as Mtb-permissive neutrophils rather than T cell-suppressive MDSCs.

Next, we considered the possibility that Mtb infection may alter the immune response to LCMV and consequently impede the control of viral infection. This effect may in turn contribute to the immunopathology in coinfected mice. Hence, we also analyzed the LCMV-specific immune response. The serum virus titers of LCMV-infected and coinfected groups after 3 days of LCMV Arm infection were similar. The virus was eliminated at 7 days post LCMV infection in both groups (Supplementary Fig. 6a). However, in contrast to the effect of type I IFN on bacterial loads, type I IFN receptor blockade increased the serum virus titer (Supplementary Fig. 6a). Both groups also showed a significant IFN-γ response to the LCMV peptides $GP_{66-80}$ (Supplementary Fig. 6b and 6c) and $GP_{33-41/276-286}$ (Supplementary Fig. 6d, e) for $CD4^+$ and $CD8^+$ T cells, respectively. LCMV-specific T cell responses were also curtailed upon IFNAR-1 blockade (Supplementary Fig. 6c, e). These data suggest that exacerbation of pulmonary pathology was triggered by type I IFN induced by LCMV Arm infection, not by LCMV Arm itself, and that the action of type I IFN signaling in host protection against bacterial and viral infections may be different.

Cognate interactions between antigen (Ag)-specific $CD4^+$ T cells and Ag/MHC complexes expressed on Ag-presenting cells are required to optimally induce Th1 responses. Since DCs activate T cells through diverse functional interactions, we also analyzed DC functional surface markers during Mtb infection. The level of MHCII molecules on DCs was reduced in LNs (Supplementary Fig. 6f, g) and pulmonary lesions (Supplementary Fig. 6h, i) of the coinfected mice. In addition to Mtb-specific T cell responses, MHCII levels on DCs were rescued by IFNAR-1 blockade in vivo (Supplementary Fig. 6g, i). Thus, downregulation of MHCII levels on DCs may be insufficient to generate optimal Th1 responses in both LNs and lungs.

**A high level of sustained type I IFNs induced by intratracheal administration of poly I:C also exacerbates TB pathology.** We detected a detrimental process of type I IFN signaling in TB pathology. To investigate whether pulmonary lesions are generated due to temporal type I IFN production induced by nonviral agents, mice were systemically exposed to poly I: C, which is a potent inducer of type I IFN responses in vivo. To induce enhanced and sustained type I IFN signaling, we intratracheally injected poly I: C into mice three times at 14, 15, and 16 days post Mtb infection (Supplementary Fig. 7a). At 26 days post Mtb infection, the pulmonary lesions were deteriorated after multiple poly I: C treatments, and this effect was dramatically alleviated by type I IFN receptor blockade (Supplementary Fig. 7b, c). In addition, the reduction in body weight in the multiple poly I:C-injected group was severe, which was not observed when the type I IFN receptor was blocked (Supplementary Fig. 7d). These results were also

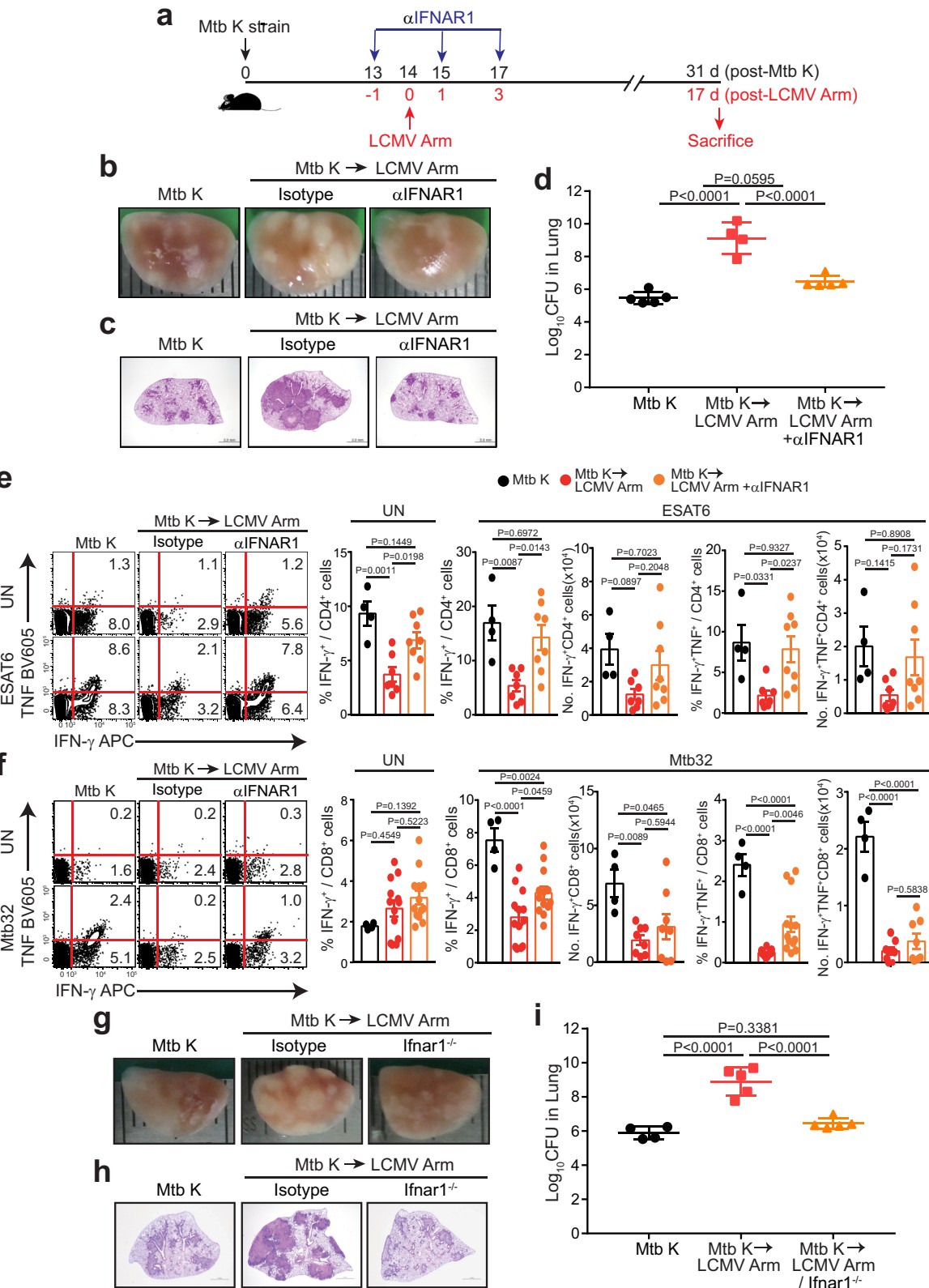

supported by changes in the bacterial loads among three groups (Supplementary Fig. 7e). In aspects of Mtb-specific T cell responses, IFN-γ-producing Mtb-specific CD4+ and CD8+ T cells were curtailed by poly I: C injection, but these reduced functionalities were reinvigorated by type I IFN receptor blockade (Supplementary Fig. 7f). Multiple poly I: C injections exacerbated pulmonary pathology, but the extent was less than

that with viral coinfection. Additionally, the IFN-α levels and their sustainability in serum and lung lysates were significantly higher in LCMV Arm-infected mice than in triple poly I: C-injected mice (Supplementary Fig. 8a, b), which was correlated with the degree of pulmonary lesion exacerbation. Hence, the degree of pathology exacerbation was dependent on type I IFN levels and their sustainability in vivo.

**Fig. 3 Beneficial effect of early type I IFN signaling blockade on exacerbated pulmonary pathology and Mtb-specific T cell responses in vivo. a** At −1, 1, and 3 days after LCMV Arm infection, a monoclonal anti-IFNAR-1 antibody (αIFNAR-1) was administered i.p. into coinfected mice. The mice were sacrificed at 31 days post Mtb infection. **b** Gross pathology or **c**, H&E staining of the lungs in each group. Scale bars, 2 mm. **d** Bacterial loads in the lungs of each group ($n = 4$ or 5). Isolated lung lymphocytes were restimulated ex vivo with **e**, an ESAT6 peptide pool for $CD4^+$ T cells ($n = 4, 6, 7, 8, 9$) or **f** an Mtb32 peptide pool for $CD8^+$ T cells ($n = 4, 8, 13$). The frequencies of IFN-γ- and TNF-producing $CD4^+$ T cells or $CD8^+$ T cells were analyzed by flow cytometry. **g**–**i** C57BL/6 or $Ifnar1^{-/-}$ mice were infected with Mtb. Some mice from each group were subsequently infected 14 days later with LCMV Arm. The mice were sacrificed at 31 days post Mtb infection. **g** Gross pathology or **h** H&E staining of the lungs in each group. Scale bars, 2 mm. **i** Bacterial loads in the lungs of each group ($n = 4$ or 5). The data were analyzed by **e**–**f**, two-tailed unpaired Student's $t$-test and **d**, **i** one-way ANOVA with *post hoc* Tukey's test. Plots show the mean ± SEM. **d**, **i**, These data are representative of at least two independent experiments. **e**–**f**, The results were pooled from two or three independent experiments. Source data are provided as a Source Data file.

**Single-cell analysis shows type I IFN-mediated downregulation of *Cxcl9/10* expression and the Th1 response in the lung during viral coinfection**. To further understand the type I IFN-mediated exacerbation of the T cell response and pathology during coinfection, we applied recently advanced single-cell transcriptome analysis technology with four different groups of mice: G1. naïve, G2. infected with Mtb alone, G3. coinfected with Mtb and LCMV, and G4. coinfected and treated with type I IFN receptor blockade (Fig. 3a). We performed both scRNA-seq and scTCR-seq for $CD45^+$ immune cells isolated from the lungs at 21 days post Mtb infection (7 days post LCMV infection) (Fig. 4a). Dimension reduction analysis of transcriptome profiles for pulmonary immune cells using uniform manifold approximation and projection (UMAP) showed 18 distinct cell clusters (c1-c18), in which both myeloid and lymphoid cells existed in considerable proportions (Fig. 4b). We identified major cell types based on classical marker gene expression for both lymphocytes and myeloid cells: B, T, $CD8^+$ T, $CD4^+$ T, regulatory T, and natural killer (NK) cells, neutrophils, and macrophages, as shown in the corresponding cell clusters (Supplementary Fig. 9a). UMAP clustering for each group of mice showed that the levels of both lymphocytes and myeloid cells significantly increased after Mtb infection. When mice were coinfected with LCMV, the proportions of some macrophage/monocyte clusters (c3 and c4) significantly increased but decreased upon type I IFN receptor blockade. Regarding T cell clusters, coinfection resulted in a tendency of CD4-1 (c6) cluster depletion and CD8-1 (c5) cluster enrichment, but the CD4-1 cluster was rescued by type I IFN receptor blockade (Fig. 4c). Some clusters expressing both myeloid markers and TCR were excluded from the analysis because they seemed to be doublets. Reconstruction of myeloid clusters during coinfection led us to investigate gene expression patterns related to inflammatory cytokines and chemokines. Given our earlier observation of type I IFN-mediated TB pathology exacerbation during coinfection, we tried to find candidate inflammatory genes whose expression was reduced upon coinfection compared to that with Mtb infection alone but rescued by type I IFN receptor blockade. Interestingly, the *Cxcl9* and *Cxcl10* genes were dominantly expressed after Mtb infection (Fig. 4d), and the Macrophage-2 (c3) cluster was the predominant source of *Cxcl9* and one of the dominant sources of *Cxcl10*, along with the Neutrophil-2 cluster among myeloid clusters (Supplementary Fig. 10a). The *Cxcl9* and *Cxcl10* expression in Macrophage-2 (c3) significantly increased after Mtb infection and decreased upon coinfection but was partially restored by type I IFN receptor signaling blockade (Fig. 4d, e). In addition to scRNA-seq analysis, we also investigated the predominant source of CXCL9 at the protein level in myeloid cell populations by flow cytometry analysis. In the Mtb K alone-infected group, $CD11b^+F4/80^+$ cells were the dominant cell type expressing CXCL9 (Supplementary Fig. 10b, c). We also found that the frequency and number of $CD11b^+F4/80^+CXCL9^+$ cells significantly decreased in the

coinfected group, but this decrease was rescued by type I IFN receptor blockade (Supplementary Fig. 10b, c).

Next, we performed UMAP dimension reduction analysis for only T cell populations to further analyze T cell gene expression along with clonotype. $CD4^+$ and $CD8^+$ T cells were subdivided into 4 and 3 clusters, respectively (Fig. 4f). Based on gene signature enrichment, one naïve and two effector populations were defined for both $CD4^+$ T and $CD8^+$ T cells (Supplementary Fig. 11a). Regulatory T cells were also identified among $CD4^+$ T cells as CD4-4 (c7). Regarding the pattern of T cell clonotypes, we observed that the proportion of IFN-γ-expressing $CD4^+$ T cells with expanded clonotypes profoundly increased post Mtb infection. However, upon coinfection, the proportion of IFN-γ-expressing expanded clonotypes of $CD4^+$ T cells dramatically decreased, and this tendency was rescued by type I IFN receptor blockade (Fig. 4g). Consistent with the UMAP data, whereas the coinfected group exhibited low numbers of expanded clonotypes in effector $CD4^+$ T cell clusters, the Mtb-only infected and type I IFN receptor blockade groups displayed an increased number of expanded clonotypes (Fig. 4h). Given our earlier flow cytometric data regarding the presence of IFN-γ-producing $CD4^+$ T cells but not $CD8^+$ T cells in the lung even without ex vivo antigen restimulation (Fig. 2h), the upregulated expression of genes related to Th1 and activation signatures in Mtb-infected mice seemed to be consistent (Fig. 4i and Supplementary Fig. 11b). Additionally, consistent with our previous data (Fig. 3e), type I IFN blockade seemingly promoted the Th1 response even in coinfected mice (Fig. 4i and Supplementary Fig. 11b). *Cx3cr1* expression in effector $CD4^+$ T cell clusters was downregulated in the coinfected group and was partially rescued by type I IFN receptor blockade (Supplementary Fig. 10d). Unlike the tendency for $CD4^+$ T cells, expression of effector molecule-encoding genes, such as *Gzmb* and *Prf1*, and effector $CD8^+$ T cell signatures increased more in coinfected mice independent of type I IFN receptor blockade (Supplementary Fig. 11c, d). The number of expanded clonotypes appeared to increase in effector $CD8^+$ T cell clusters of both coinfected groups compared with those of the Mtb alone-infected group (Supplementary Fig. 11e). Because LCMV infection induces effector $CD8^+$ T cell expansion more rapidly than $CD4^+$ T cell expansion at early time points, both the coinfected and type I IFN receptor blockade groups exhibited similar $CD8^+$ T cell phenotypes. Furthermore, since the enhanced and similar phenotypes of the C4-1 (c2) cluster were commonly exhibited in both the Mtb alone-infected and type I IFN receptor blockade groups, with alleviated pulmonary lesions, the proportion of effector clusters of $CD4^+$ T cells may be important for TB protection in vivo. Overall, the single-cell analysis of pulmonary immune cells provided some clues, such as decreased CXCL9/10 expression and Th1 responses during coinfection, to explain type I IFN-mediated pathology.

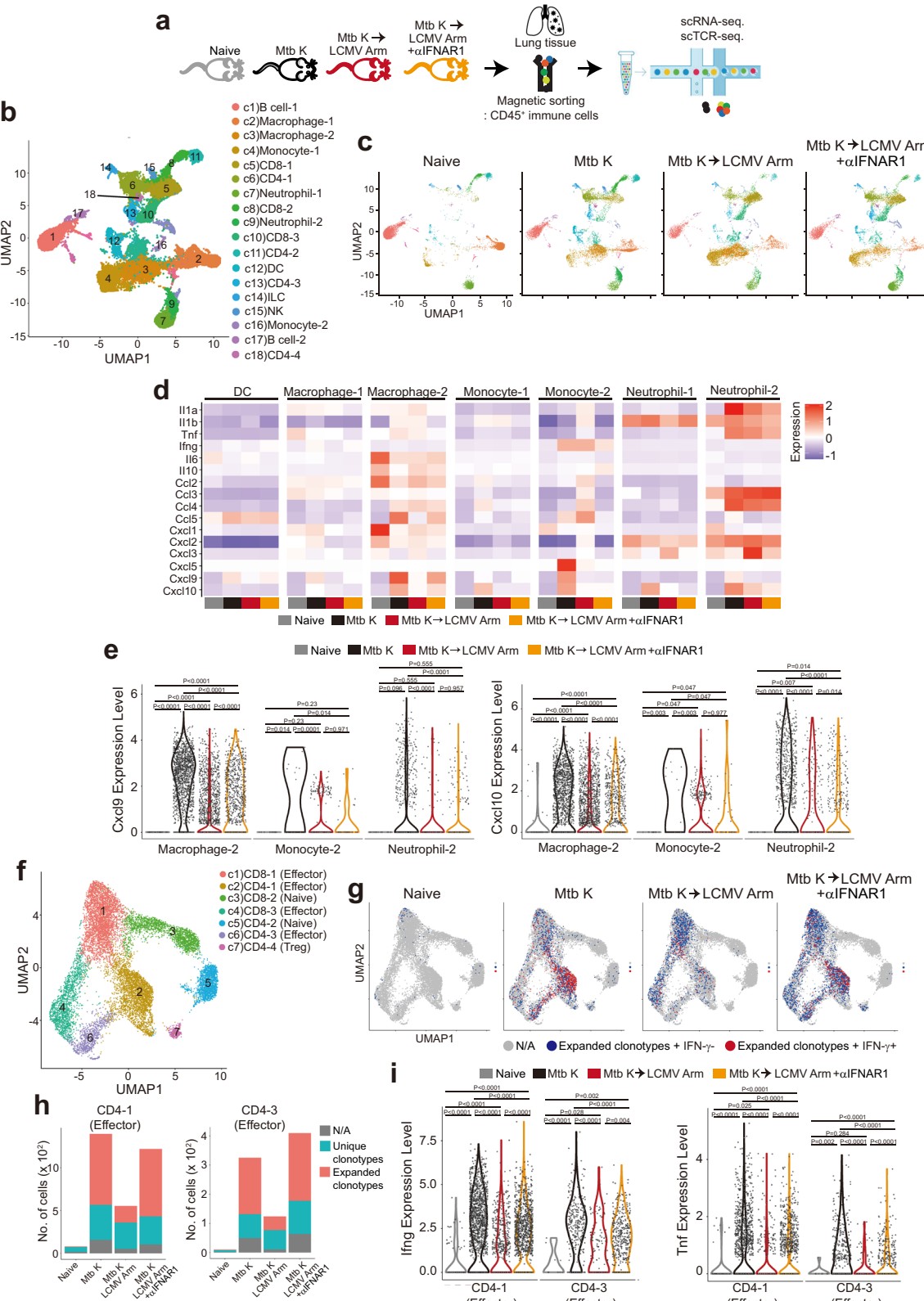

**Type I IFN receptor signaling significantly inhibits CXCL9 protein expression in the lung upon virus infection of Mtb-infected mice.** Since coinfection led to reduced *Cxcl9/10* transcription in one subset of pulmonary macrophages, which was restored by type I IFN receptor blockade, we examined whether the protein expression level would also show a similar pattern in pulmonary macrophages by using multiplex immunofluorescence

staining. In the inflammatory region of the lungs of Mtb-infected mice, CXCL9 was detected at a similar level between the Mtb alone-infected and type I IFN receptor blockade groups, but the CXCL9 expression levels and signal intensities were the lowest in the coinfected group (Fig. 5a, c). Additionally, in noninflammatory regions, CXCL9 was rarely expressed in coinfected mice, and this pattern was rescued by type I IFN receptor

**Fig. 4 Pulmonary immune cell landscape and cell state remodeling by type I IFNs produced in response to viral infection during Mtb infection. a** Single-cell suspensions of CD45$^+$ pulmonary immune cells from the indicated 4 groups at 21 days post Mtb infection were analyzed by scRNA-seq or scTCR-seq as described in the STAR methods. **b** UMAP plots of cells from all scRNA-seq samples, colored according to cell type. **c** UMAP plots of all scRNA-seq clusters among the naïve, Mtb alone-infected, coinfected, and type I IFN receptor blockade groups. **d** Heatmap of normalized expression of selected inflammatory cytokine and chemokine genes in each myeloid cluster. **e** Violin plots showing the expression of the indicated marker gene in the indicated myeloid clusters. **f** UMAP plots of T cells across all samples, colored according to identified clusters. **g** UMAP plots of expanded clonotypes and Ifng-expressing cells in T cell clusters. **h** The number of unique or expanded clonotype cells in effector CD4$^+$ T cell clusters is summarized in the bar graph. **i** Violin pots showing the expression of the indicated marker gene from the indicated T cell clusters. N/A indicates "not available" populations, which means that the gray bar represents the nonunique or expanded clonotypes. The CD45$^+$ immune cells were pooled from mice ($n = 3$) in each group. **e, i** two way with Wilcoxon rank-sum test. The data were analyzed by Wilcoxon rank sum test.

blockade (Fig. 5a, c). The CXCL9 protein was visible mostly in the cytoplasm (Fig. 5b). Then, we investigated the cellular source of CXCL9 in Mtb-infected pulmonary lesions. Generally, CXCL9 is secreted by either immune cells, such as macrophages and monocytes, or by nonimmune cells, including endothelial and epithelial cells[43,44]. Lung tissue staining with multiple antibodies against CXCL9, CD11b, CD31, and PDPN showed that PDPN$^+$ lung epithelial cells appeared to express CXCL9 in the noninflammatory regions, while CD11b$^+$ cells mainly expressed CXCL9 in the inflammatory region (Fig. 5d). Collectively, our data suggest that type I IFNs exacerbate pulmonary pathology by inhibiting the migration of Mtb-specific T cells to pulmonary lesions via the regulation of CXCL9 expression in a type I IFN-dependent manner.

**Type I IFN suppresses the proliferation and functionality of Mtb-specific T cells in vivo.** Along with the negative effects of type I IFN induced by LCMV Arm infection in TB pathogenesis depicted above, we next investigated whether the effect of type I IFN on CD4$^+$ T cell proliferation and functionality occurred in an Mtb Ag-specific manner in vivo. For this purpose, we purified CD4$^+$ T cells from naïve P25-Tg mice (Ly5.1$^+$), engineered them to express a TCR specific for the immunodominant Mtb antigen Ag85B[45], and adoptively transferred them to naïve mice (Ly5.2$^+$). At 1 day after adoptive transfer, the mice were infected with Mtb and subsequently infected with LCMV Arm 14 days after Mtb infection (Fig. 6a). The frequency and number of P25 cells in the lungs were reduced in coinfected mice, and type I IFN signaling blockade rescued the frequency of total P25 cells (Fig. 6b). In addition, we further analyzed the kinetics of P25 cells in the lungs. The frequency and number of P25 cells at 3 weeks post Mtb infection were higher than those at 2 weeks post Mtb infection, which suggested that a major influx and accumulation of P25 cells occurred between 2 and 3 weeks after Mtb infection (Supplementary Fig. 12a). This accumulation tendency would fit our overall model of coinfection with LCMV starting at 3 weeks post Mtb infection, showing no impact on pathology and indicating that establishment of robust Th1 responses could alleviate coinfection-mediated pulmonary pathology and bacterial outgrowth (Supplementary Fig. 3h–l). Similar to the total P25 cells in the lung, the frequency of Ki-67$^+$ cells among P25 cells was also reduced in coinfected mice and increased upon type I IFN signaling blockade (Fig. 6c). The frequency of CD44$^+$ cells among P25 cells, which is the activation marker of T cells, was decreased in coinfected mice and rescued by type I IFN signaling blockade (Fig. 6c). Given our earlier data regarding type I IFN-dependent regulation of CXCL9 and CXCL10 (Figs. 4e and 5c), we checked the expression of their chemokine receptor CXCR3 and T-bet, a Th1-associated transcription factor. There was no significant difference between three groups, similar to the expression of Ki-67 and CD44 in P25 cells, although CXCR3 expression slightly decreased in the coinfected group, which tended to be rescued by type I IFN receptor blockade (Supplementary Fig. 12b). We also examined the function of P25

cells ex vivo. For this purpose, we harvested the lung, and suspended single cells from the infection site were stimulated with Ag85B peptides. At 21 days post Mtb infection, P25 cells from the Mtb-infected group produced IFN-γ and TNF at abundant levels, whereas the function of P25 cells in coinfected animals dramatically decreased. This curtailed activity was regained upon type I IFN signaling blockade (Fig. 6d). Additionally, the lung bacterial loads were inversely correlated with the frequency and number of P25 cells or IFN-γ$^+$ P25 cells in the lung, which suggests that Mtb-specific T cells and their functions in the lungs play a important function of host protection from pulmonary pathology induced by viral coinfection (Fig. 6e and Supplementary Fig. 12d). Taken together, these data demonstrate that type I IFN suppresses the proliferation and function of Mtb-specific T cells in vivo and that the dampened Th1 immune response contributes to pulmonary pathology exacerbation.

**Decreased migration of Mtb-specific T cells from dLNs to lung tissue is related to diminished levels of CXCL9 and CXCL10 induced by type I IFN.** To elucidate the timing-specific negative function of type I IFN signaling in the remarkable downregulation of Mtb-specific Th1 responses in the coinfection model, we next examined an initial event in the dLN, which is the first location of Mtb-specific T cell generation[46]. We found that the frequency and number of P25 cells in the lungs significantly decreased in coinfected mice compared to those in Mtb alone-infected mice, whereas those in the dLNs were not different in these two groups of mice (Fig. 7a, b). Interestingly, the reduced frequency and number of P25 cells in the lung of coinfected mice was rescued by type I IFN receptor blockade, suggesting that the pulmonary migration of Mtb-specific CD4$^+$ T cells was hindered by type I IFN signaling in the coinfected group. Along with the normal frequency of P25 cells in LNs, in our scRNA-seq data, the expression levels of *Cxcl9* and *Cxcl10* in the lung tended to be downregulated in the coinfected group. CXCL9 and CXCL10 are major chemokines that facilitate the migration of T cells to their focal sites[47]. Hence, we hypothesize that the migration of Mtb-specific T cells from LNs to the pulmonary infection site is hindered by type I IFN signaling downregulating local CXCL9 and CXCL10 expression. Therefore, we analyzed the in vivo levels of chemokines in the lungs. At 7 days post LCMV infection, the protein levels of CXCL9 and CXCL10 in the lungs of coinfected mice were lower than those in the lungs of Mtb-infected mice. These chemokine levels increased with type I IFN receptor blockade (Fig. 7c, d). *Cxcl9* and *Cxcl10* mRNA levels decreased in the coinfected group, and the *Cxcl9* mRNA expression level significantly increased upon type I IFN receptor blockade (Fig. 7c, d). Additionally, the CXCL9 and CXCL10 levels were positively correlated with the frequencies of P25 cells and IFN-γ$^+$ P25 cells in the lung (Fig. 7e, f). Moreover, the levels of these chemokines were significantly negatively correlated with bacterial loads (Fig. 7g), supporting the hypothesis that the chemokines are significantly involved in Mtb-specific CD4$^+$ T cell recruitment to

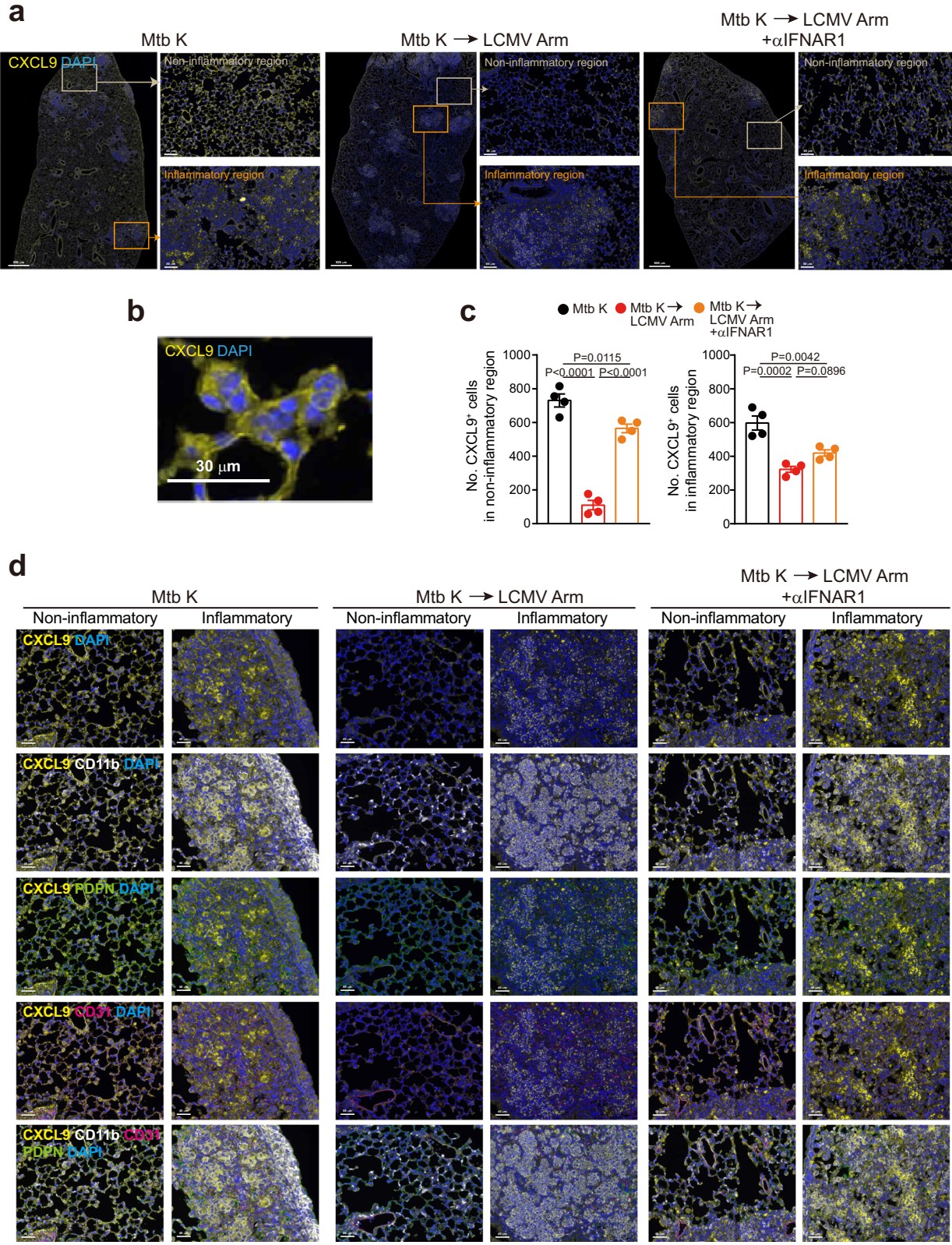

**Fig. 5 Rescued CXCL9 expression through type I IFN receptor blockade in pulmonary regions.** Seven days post LCMV infection, we analyzed CXCL9 expression in the lung by multiplex immunofluorescence staining. **a** Representative images of CXCL9 (yellow) immunofluorescence and DAPI (blue) staining in noninflammatory or inflammatory regions of lung tissues. Scale bars, 800 μm or 50 μm. **b** Representative images of CXCL9 and DAPI staining in lung tissues. Scale bars, 30 μm. **c** The number of CXCL9+ cells in each randomly selected region was counted and is summarized in the bar graphs (n = 4). One-way ANOVA with *post hoc* Tukey's test. Plots show the mean ± SEM. **d** Representative images of CXCL9 (yellow), CD11b (white), CD31 (magenta), and PDPN (green) immunofluorescence and DAPI (blue) staining of lung tissues. Scale bars, 40 μm. The data are representative of single experiments. Source data are provided as a Source Data file.

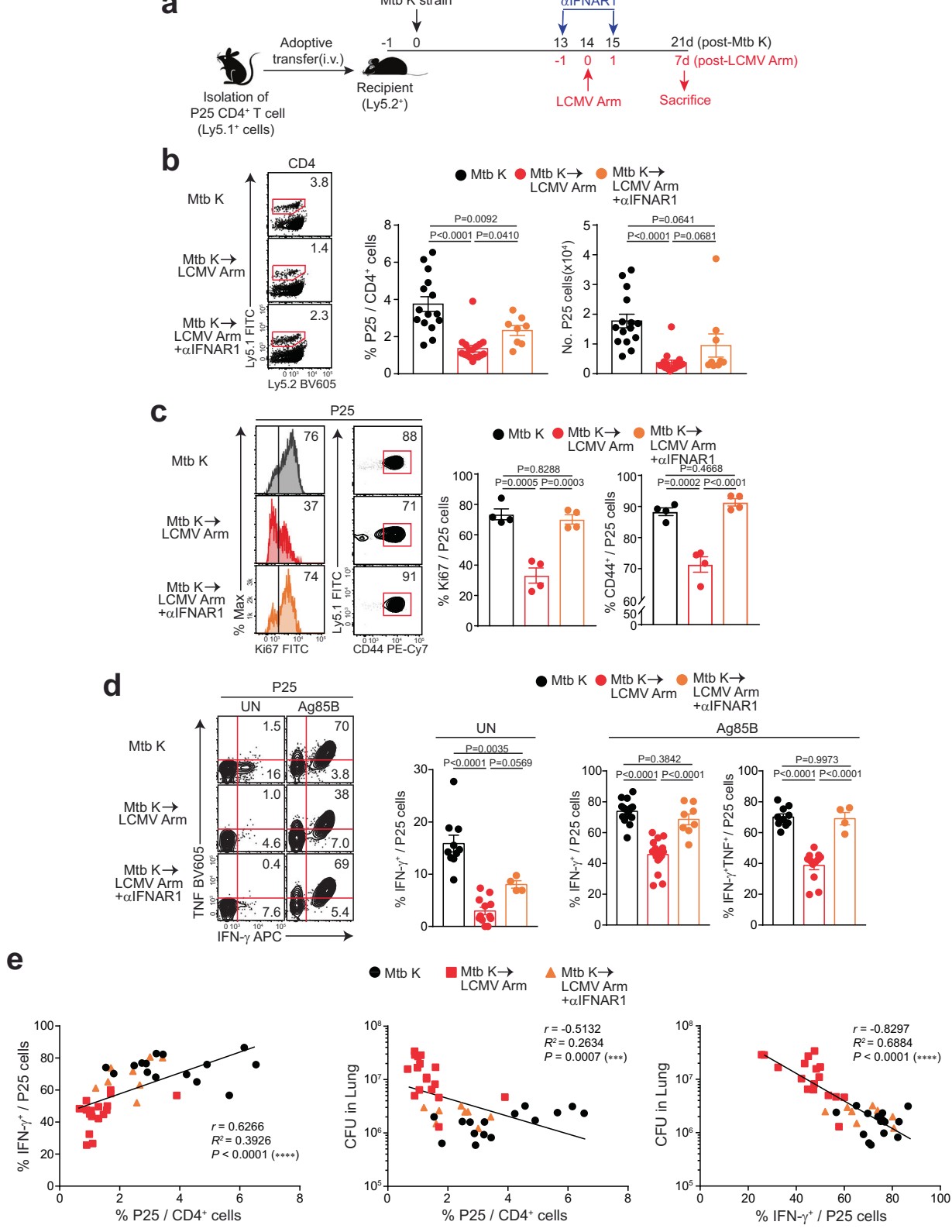

pulmonary lesions and contribute to uncontrolled Mtb growth in vivo. Furthermore, in our results, when IFN-γ was predominantly produced during Mtb infection before viral coinfection, the host was protected against type I IFN-induced immunopathology (Supplementary Fig. 3). Thus, we hypothesized that there may be counterregulation between type I IFNs and IFN-γ on CXCL9/10 expression. To validate the regulation of

CXCL9 and CXCL10 expression by type I and type II IFNs in vitro, we treated RAW264.7 cells, a monocyte/macrophage mouse cell line, with type I and type II IFNs. The CXCL9 level was increased by IFN-γ treatment in a dose-dependent manner. When RAW264.7 cells were cotreated or pretreated with type I IFNs in the presence of IFN-γ, the CXCL9 expression was significantly decreased (Fig. 7h, i). However, when RAW264.7 cells

**Fig. 6 Suppressed Mtb-specific T cell proliferation and function by type I IFN signaling. a** Naïve CD4[+] T cells were isolated from the spleens of P25 mice (Ly5.1[+]) and adoptively transferred via i.v. injection to C57BL/6 naïve mice (Ly5.2[+]). At 1 day after transfer, the mice were infected with Mtb. Some mice from each group were subsequently infected 14 days later with LCMV Arm. At -1 and 1 day after LCMV Arm infection, anti-IFNAR-1 antibodies were administered i.p. into coinfected mice. The mice were sacrificed at 21 days post Mtb infection. **b–d,** Ly5.1[+] donor P25 cells in the lung were analyzed by flow cytometry. **b** The frequency and number of Ly5.1[+] P25 cells among CD4[+] T cells. Numbers in the plots indicate the percentage of Ly5.1[+] transferred cells among CD4[+] T cells (*n* = 8, 9, 15, 16). **c** The frequencies of Ki-67[+] and CD44[+] cells among Ly5.1[+] transferred cells were analyzed by flow cytometry and are summarized in the graph. Numbers in the plots indicate the percentage of Ki-67[+] or CD44[+] cells among Ly5.1[+]-transferred cells (*n* = 4). **d,** Isolated lung lymphocytes were restimulated ex vivo with Ag85B peptides, stained, and assessed for Ly5.1[+] cells among CD4[+] T cells (*n* = 4, 8, 10, 12, 15, 17). **e** The correlation between the frequency of P25 cells or IFN-γ[+] P25 cells in the lung and bacterial loads was analyzed (*n* = 8, 15, 17). The data were analyzed by **b–d**, one-way ANOVA with *post hoc* Tukey's test or **e**, two-way test with Pearson's correlation. Plots show the mean ± SEM. **b, d, e,** The data are pooled from two or three independent experiments or **c**, are representative of a single experiment. Source data are provided as a Source Data file.

were pretreated with IFN-γ and then type I IFNs were added, CXCL9 expression was not affected by type I IFNs (Fig. 7j), indicating that the CXCL9 regulation was dependent on the sequence of type I and type II IFN signaling. In addition, the BCG-vaccinated group, which showed ameliorated exacerbation of pulmonary pathology induced by viral coinfection, already exhibited enhanced expression of CXCL9 at the time of LCMV Arm coinfection (14 days post Mtb infection) (Supplementary Fig. 13a, b). These data together with the result showing a higher Mtb-specific Th1 immune response in the lungs of the BCG-vaccinated mice (Supplementary Fig. 3a–g) suggest that accelerated generation of pulmonary Mtb-specific Th1 cells post Mtb infection in BCG-vaccinated mice, compared to that in unvaccinated mice, could contribute to IFN-γ-induced CXCL9 production in the lung, which subsequently leads to rapid accumulation of Mtb-specific Th1 cells that are sufficient to control bacterial outgrowth and pulmonary pathogenesis. Furthermore, when mice were injected with multiple poly I: C, they exhibited the reduced expression of CXCL9 in CD11b[+]F4/80[+] populations and this curtailed expression of CXCL9 was rescued by type I IFN receptor blockade (Supplementary Fig 13c and 13d), which suggest that type I IFN induced by poly I: C injection inhibited the expression of CXCL9 in vivo. In contrast, CXCL10 was expressed in response to both type I IFNs and IFN-γ without additional effects or signaling sequences (Fig. 7h–j). In summary, these data suggest that type II IFN is required for the expression of CXCL9 and CXCL10 in lung infection sites, which play a critical function of Mtb-specific Th1 cell migration from LNs, and that type I IFNs induced by viral coinfection may obstruct this process when type I IFN signal transmission precedes IFN-γ signaling.

## Discussion

The risk of TB-associated mortality due to coinfection with other infectious agents has been increasingly reported and has received greater attention since the emergence of HIV and COVID-19[48]. Nevertheless, our understanding of the underlying mechanisms in coinfection with Mtb and viruses remains insufficient, particularly regarding T cell responses.

A very recent study elucidated the immunological mechanism underlying exacerbation of Mtb pathogenesis by chronic viral coinfection. In the model, mice were first infected with virus, and Mtb was then coinfected. The authors suggested that increased TNF levels induced by virus and coinfection hindered DC-mediated Mtb Ag delivery to LNs and increased Mtb replication along with redirected CD4[+] T cell differentiation from Th1 to Th17-type T cells, which led to worse pulmonary inflammation[49]. Intriguingly, in our coinfection model, mice were infected in opposite order, resulting in a type I IFN-CXCL9 axis during Mtb and virus coinfection being a major host determinant of the exacerbation of Mtb infection pathogenesis. Additionally, we found that TNF blockade did not support improvements in TB pathogenesis in our coinfection model. Thus, notably, the

immunopathology of TB with viral coinfection immunologically differs according to the infection order. Nevertheless, both studies identified that curtailed CD4[+] T cell responses in Mtb-infected lungs are a main cause of the progression to severe TB regardless of the viral coinfection order.

Type I IFNs were also recently identified as potential mediators of the pathology induced by other types of infectious agents, including mycobacteria[50]. Although positive effects of type I IFNs on immune cell activation have been reported in several earlier in vitro Mtb infection studies[51,52], type I IFNs exacerbate pulmonary pathology in vivo. In mouse Mtb infection studies, IFNAR-1-deficient mice exhibited reduced bacterial loads and enhanced host survival compared to those of WT mice[53,54]. Moreover, when Mtb-infected mice were treated with poly-ICLC to stimulate type I IFN production at a high level via TLR3 activation, significant increases in lung bacterial loads together with necrotic granulomas were observed, but this phenomenon did not occur in mice lacking IFNAR-1[37]. The authors explained the severe TB progression in their model as the counter-regulation between two inflammatory cytokines, IL-1 and type I IFN, through the balance of PGE2 production. However, the effect of type I IFNs on Mtb-specific T cells remains unclear. Mtb-specific T cells were shown to be activated by antigen-presenting cells in pulmonary dLNs and to migrate to pulmonary infection sites at 2 weeks post Mtb infection[55]. We demonstrated that type I IFNs led to retention of Mtb-specific CD4[+] T cells in LNs and delayed their migration into the pulmonary infection site. However, in our coinfection model, it remains unclear whether type I IFN signaling is acting directly or indirectly on the T cell response. It is well known that both innate and adaptive immune cells are affected by type I IFN signaling during bacterial or viral infection because both cell types ubiquitously express type I IFN receptor in their surface[50]. According to our current study, it is conceivable that type I IFN signaling might downregulate IFN-γ-mediated expression of CXCL9 and CXCL10 in innate immune cells, thereby hindering the migration of Mtb-specific T cells into the lung and finally leading to severe TB pathogenesis. In addition to this possibility, T cells also could be directly altered by type I IFN signaling. In a previous study, type I IFN signaling had a direct influence on the clonal expansion of CD4[+] T cells during LCMV infection[56]. Furthermore, another report suggested that effects of type I IFN to CD8[+] T cells may depend on the timing of type I IFN exposure. For example, if CD8[+] T cells are exposed to type I IFN signaling before antigen encounter, type I IFNs would be harmful to T cell proliferation. However, if they are exposed simultaneously to antigen and type I IFNs, type I IFN signaling positively acted on T cell proliferation[57]. Since the direct or indirect effect of type I IFNs on Mtb-specific T cells during TB is still unclear, thus, further studies should be conducted to determine whether the direct action of type I IFN on Mtb-specific T cell responses will affect pathological exacerbation of TB. Furthermore, by using IFNAR-1 blockade and complementary

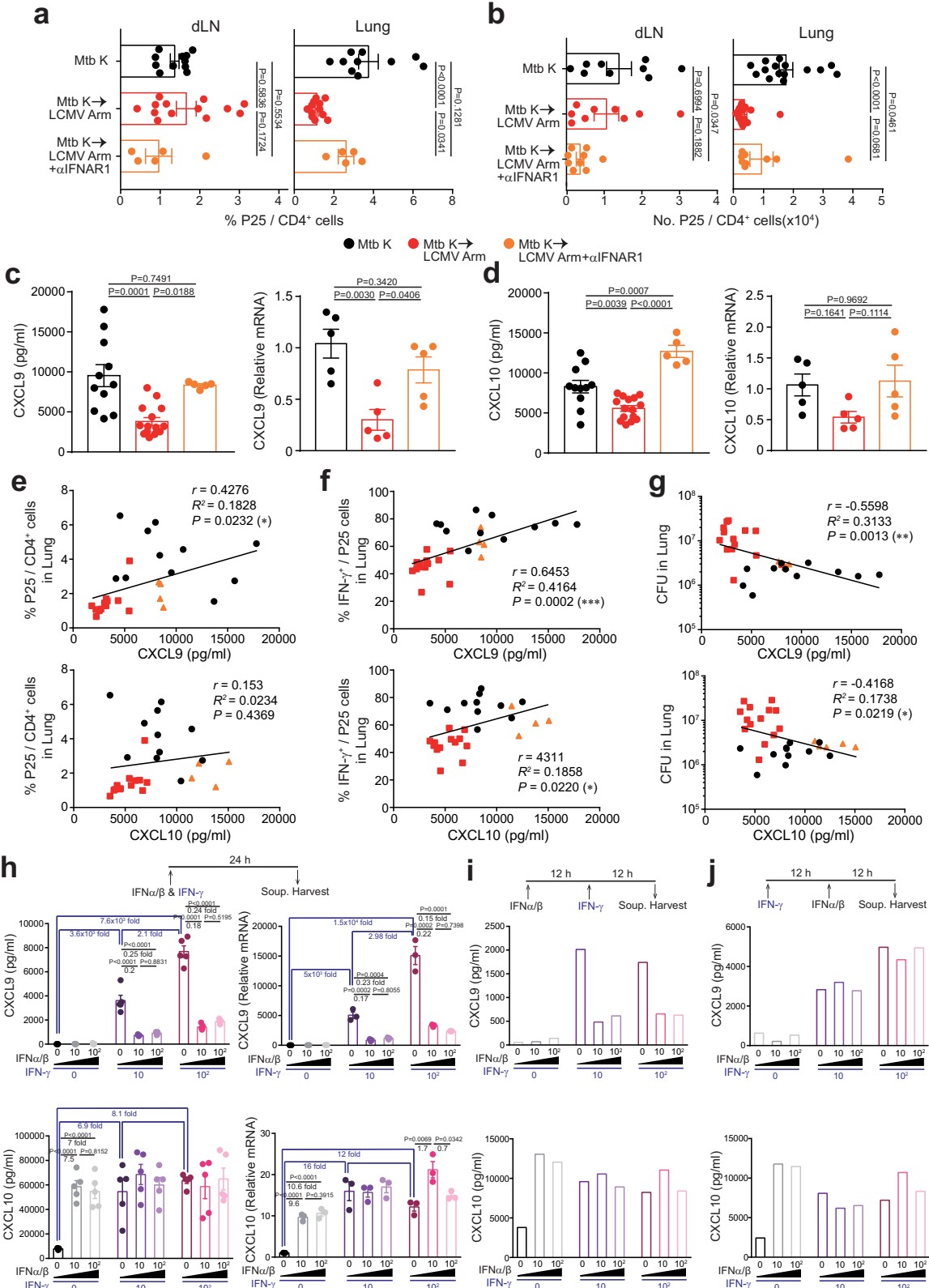

application of type I IFN receptor knockout mice, we clearly show that much of the exacerbated pathology caused by coinfection is due to type I IFN signaling. However, Mtb-specific T cell responses or expression of CXCL9 and CXCL10 was not fully restored by IFNAR-1 blockade, which suggests that some type I IFN-independent mechanisms stimulated by LCMV infection may also contribute to exacerbated pulmonary pathology in coinfected mice. Since the risk of TB-associated mortality due to coinfection with other infectious agents has been increasingly reported, further research should be conducted with diverse mouse coinfection models.

In contrast to type I IFNs, IFN-γ has a central function for protecting the host from bacterial infections. Regarding the balance between type I IFNs and IFN-γ, it has been reported that type I

**Fig. 7 Curtailed migration of Mtb-specific T cells by negative regulation of chemokines by type I IFN. a** The frequency and **b**, number of Ly5.1[+] donor P25 cells in the dLNs and lungs were analyzed by flow cytometry and are summarized in the graph. **a** (n = 4, 5, 10, 11, 12, 13) or **b**, (n = 8, 9, 14, 15). At day 7 post LCMV Arm infection, **c** CXCL9 or **d** CXCL10 protein and RNA levels in lung homogenates were analyzed by ELISA and qRT-PCR, respectively (n = 5, 11, 15). The correlations between chemokine levels and **e**, the frequency of P25 cells in the lung (n = 4, 11, 13), **f**, IFN-γ[+] cells in the lung (n = 4, 11, 13) and **g**, bacterial loads were analyzed (n = 5, 11, 14). **h–j** The CXCL9 or CXCL10 protein and RNA levels in the culture supernatant of RAW264.7 cells in the indicated groups treated with type I IFNs and type II IFN. The data were analyzed by **a–d**, **h**, one-way ANOVA with *post hoc* Tukey's test or **e–g**, two-way test with Pearson's correlation. Plots show the mean ± SEM. **a–g**, The data are pooled from two independent experiments or **h**, representative of at least two independent experiments (n = 3, 5 wells/group) or **i**, **j**, representative of a single experiment. Source data are provided as a Source Data file.

IFNs exert a negative effect, downregulating the IFN-γ responsiveness of innate immune cells, such as macrophages and monocytes[58,59]. Additionally, during Mtb infection, IFN-γ-induced nitric oxide synthase and IL12/24 p24 production by pulmonary myeloid cells was curtailed by type I IFNs in vivo[60]. However, the effect of the timing of the production of IFN-γ and type I IFNs on host immunity in vivo has not been fully elucidated. Here, we showed that viral coinfection-induced exacerbated pulmonary pathology and uncontrolled Mtb growth due to the action of type I IFNs were prevented when mice were immunized with BCG or infected with LCMV at 3 weeks post Mtb infection, and both groups had Mtb-specific IFN-γ-producing T cells that already existed to some extent in the lung. Therefore, the timely balance between the amounts of type I IFNs and the preexistence of Mtb-specific IFN-γ-producing T cells in the lung might be critical determinants of exacerbated immunopathology in vivo.

Because scRNA-seq can define the transcriptomic heterogeneity of complex immune cell populations and thus identify immune cell subsets in an unbiased manner, it is well suited for the study of complex inflammatory diseases. Indeed, Cai et al. performed scRNA-seq using human PBMCs from healthy controls and individuals with latent or active tuberculosis[61]. Additionally, Esaulova et al. recently described the immune landscape of TB latency and disease in the lungs of nonhuman primates, defining the features of pulmonary active TB, such as the accumulation of IFN producers and responders in the lung[62]. Here, single-cell analysis provided clues to discover the mechanism by which viral infection disrupts immune responses and exacerbates Mtb-mediated pathology. In-depth and unbiased profiling of pulmonary immune cell populations led us to discover that *Cxcl9/10* expression in pulmonary macrophages is enhanced upon coinfection but not after type I IFN receptor blockade, which was also validated with multiplex immunofluorescence analysis and ELISA. Consistent with our findings, Esaulova et al. also suggested that both type I and II IFN signals accumulate in pulmonary active disease, but they could not identify the events thereafter[62]. In comparison, we further tracked the next events after increased CXCL9/10 expression. To end this, it was necessary to use scTCR-seq because CXCL9/10 recruits effector T cells into inflamed tissues, and actively proliferating clones are easily tracked by scTCR-seq. As expected, expanded clones of CD4[+] T cells producing IFN-γ were substantially detected in the Mtb infection-only control group and type I IFN blockade coinfection group but not in the coinfection group. Therefore, the single-cell technology-based experimental findings led to the hypothesis that the relationship between type I and II IFNs may play an essential function for the pulmonary migration of Mtb-specific CD4[+] T cells.

In general, earlier infiltration of Ag-specific T cells to the infection site is important for the protection of the host against pathogens. Although a delayed T cell response is a signature of Mtb infection, rapid migration of Ag-specific T cells to the pulmonary site is mostly important for protection against Mtb[63,64] and is predominantly mediated by the chemokine and chemokine receptor axis[65]. The chemokines CXCL9 and CXCL10, which bind to the chemokine receptor CXCR3, egress T cells from

lymphoid organs to the inflammation site[66]. Although CXCL9 and CXCL10 are well known to be important for the recruitment of T cells to inflamed sites, the functions of these chemokines during Mtb infection are still unclear and controversial. In a mouse model of TB, lung-resident IL-17[+] cells were shown to be critical for the induction of CXCL9 and CXCL10, which were required for the accumulation of IFN-γ[+] memory T cells in vivo[67]. In contrast, in humans, plasma CXCL9 levels were reported to correlate with TB disease severity[68,69]. Here, we highlighted the decreased CXCL9 and CXCL10 levels in coinfected mice, resulting in depletion of Ag-specific T cells in the lung along with retention of those T cells in the dLNs of coinfected mice. Therefore, considering the previously described functions of CXCL9 and CXCL10 along with our results, systemically increased CXCL9 levels detected in human plasma might be the result of severe pulmonary pathology, not the cause of the local infected lesions. Regarding expression regulation, CXCL10 production is generally induced strongly by IFN-γ and partly by type I IFN. In contrast, CXCL9 is predominantly induced by IFN-γ and not type I IFN[70–72]. In line with these previous reports on the regulation of CXCL9 and CXCL10, in our experiments with RAW264.7 cells, CXCL9 was induced by IFN-γ but not by type I IFNs in vitro. Surprisingly, we found that type I IFNs directly inhibited CXCL9 expression induced by IFN-γ. Therefore, it is conceivable that the decrease in CXCL9 levels in the lungs of coinfected mice was due to directly elevated type I IFN levels induced by viral infection followed by perturbation of Mtb-specific T cell migration from the LN to the pulmonary infection site. Furthermore, although CXCL10 gene is well known as a type I IFN-stimulated gene, it was somewhat surprising that its expression was elevated in the lung after type I IFN signaling blockade in our coinfection model. However, given that type I IFN receptor blockade increased IFN-γ levels of T cells in the lung of coinfected mice, it is possible that the effect of the increased IFN-γ production of T cells for CXCL10 production is greater than that of reduced type I IFN signaling, resulting in an increase in CXCL10 expression in the lung. In addition, the type I IFN and IFN-γ signaling pathways has been known to share the phosphorylated-STAT1 transcription factor. Therefore, since the type I IFN signaling was inhibited by type I IFN receptor blockade, most of phosphorylated-STAT1 would be used to the strengthen IFN-γ signaling pathway for CXCL10 expression.

Although an early protective function of neutrophils has been reported[73,74], neutrophilia in TB-susceptible models is a common trait aggravating TB pathogenesis[75,76]. However, the function of neutrophils in TB immunopathology remains controversial: via T cell suppression via MDSCs versus creating a permissive intracellular niche for Mtb growth. Additionally, it has been reported that neutrophils had an MDSC phenotype and suppressive function against T cells in TB-susceptible 129S2 mice[41]. Recently, neutrophils were shown to contribute to pulmonary disease through type I IFN-induced neutrophil extracellular trap formation[38]. In our mouse infection model, we detected higher levels of neutrophil infiltration and higher Mtb numbers in lung neutrophils from the coinfected group than in those from the

Mtb-infected group. However, mice had similar bacillary loads and pulmonary pathology despite attenuated TB pathogenesis upon depletion of granulocyte populations, and similar to tumor-associated granulocytes, neutrophils had no MDSC function in vitro, indicating that the elevated neutrophil levels upon coinfection might be a consequence of the Mtb-permissive cell type, not the causative factor via MDSC suppression of Mtb-specific T cell responses.

In summary, viral coinfection during Mtb infection led to TB immunopathogenesis by downregulating the expression of the T cell migratory chemokines CXCL9/10 in infected lung tissues. Immunopathological characteristics were ameliorated by blocking type I IFN signaling but not by neutrophil or other inflammatory cytokine depletion, suggesting that type I IFNs are a key negative factor delaying the Mtb-specific Th1 cell response in the lung. Type I IFNs produced in response to viral infection affected the expression levels of factors related to T cell migration from the LN to pulmonary infection sites, such as CXCL9 and CXCL10, eventually delaying the Mtb-specific Th1 cell response in the lung and thereby inducing exacerbation of pulmonary pathology with uncontrolled bacterial growth. Thus, our current study has shown another negative function of type I IFN signaling and/or virus coinfection during Mtb infection, and reinforcing the early and timely Mtb-specific T cell responses in the lung might be a useful strategy to protect the host from viral coinfection or type I IFN-driven inflammatory tissue damage in TB.

## Methods

**Mice**. Specific pathogen-free female C57BL/6 (B6) mice (The Jackson Laboratory, JAX:000664) at 5 to 6 weeks of age were purchased from Japan SLC, Inc., and maintained under barrier conditions in a biosafety level 3 (BSL-3) biohazard animal room at the Medical Research Center of Yonsei University College of Medicine and B6 mice at 6 to 8 weeks of age were used for experiments. P25 (C57BL/6-Tg(H-2Kb-Tcra,Tcrb)P25Ktk/j)(The Jackson Laboratory, JAX:011005) mice were purchased from Jackson Laboratory (Bar Harbor, ME)[45]. $Ifnar1^{-/-}$ mice (B6.129S2-Ifnar1t-m1Agt/Mmjax) (The Jackson Laboratory, JAX:32045) were kindly gifted from Dr. Heung Kyu Lee (KAIST, Daejun, Republic of Korea). CD45.1 mice (B6.SJL-Ptprca Pep3b/BoyJ) (The Jackson Laboratory, JAX:002014) were obtained from laboratory of Young-Chul Sung (POSTECH, Republic of Korea). For experiments, each female mice at 6 to 8 weeks of age were used and background strain of all mice used is C57BL/6 (B6) mice (The Jackson Laboratory, JAX:000664). Animal maintenance and procedures were performed with approval of the IACUC of Yonsei University College of Medicine (permit number: 2016-0305). All animal studies were performed in accordance with Korean Food and Drug Administration (KFDA) guidelines. The mice were housed at temperature ($23 \pm 3°C$), humidity range 40–60, and 12 h light-dark cycle (7 am on and 7 pm off).

**Preparations of Mtb strains**. Mtb H37Rv (ATCC 27294) and H37Ra (ATCC 25177) were purchased from the American Type Culture Collection (ATCC) (Manassas, VA, USA). The Mtb K strain (from the Beijing lineage, which is predominant in Korea[77]) was obtained from the strain collection of the Korean Institute of Tuberculosis (Osong, Chungchungbuk-do, Republic of Korea). The Mtb HN878, Erdman and M2 strains were obtained from the strain collection of the International Tuberculosis Research Center (Changwon, Gyeongsangnam-do, Republic of Korea). The BCG vaccine (Pasteur 1173P2) was kindly provided by Dr. Brosch from the Pasteur Institute (Paris, France). All Mtb strains used in this study were cultured and prepared as previously described[78].

**Infection, counts, and titration**. For bacterial infections, mice were exposed to Mtb strains in the calibrated inhalation chamber of an airborne infection apparatus for 60 min, delivering a predetermined dose (Glas-Col, Terre Haute, IN, USA) of approximately 130 to 230 viable bacteria. To enumerate the bacteria, serial dilutions of organ homogenates were plated onto Middlebrook 7H11 agar plates (Becton Dickinson, San Diego, CA, USA) supplemented with amphotericin B, 2 mg/mL 2-thiophenecarboxylic acid hydrazide (Millipore Sigma), and 10% oleic albumin dextrose catalase (Difco Laboratories, Detroit, MI, USA). After incubation at 37 °C for 3 to 4 weeks, bacterial colonies were counted, and the results are presented as the mean $\log_{10}$CFU per organ.

For viral infections, mice were infected intraperitoneally (i.p.) with $1 \times 10^5$ or $2 \times 10^5$ PFU of LCMV Arm 53b, a triple-plaque-purified isolate of LCMV ARM CA1371, or intravenously (i.v.) with $2 \times 10^6$ PFU of LCMV CL13, a variant derived from an LCMV ARM CA1371 carrier mouse. The viruses were obtained from Rafi Ahmed (Emory Vaccine Center, Atlanta).

For virus titration, three to five drops of blood were collected from the retro-orbital sinus at the indicated time points, and the serum was immediately stored at −70 °C. Viral titers of serum samples were determined by plaque assay on Vero cells. Briefly, $3.5 \times 10^5$ Vero cells were plated in 35-mm wells in 6-well dishes (Costar), and the plates were incubated at 37 °C. When the monolayers were confluent, the medium was removed, and serum samples (total volume of 200 μL) were added to the cells. After absorption for 1 h at 37 °C, the cells were overlaid with 4 mL of 1% agarose (SeaKem GTG agarose; Lonza) in Medium 199 (Gibco Laboratories) supplemented with 10% FBS, penicillin/streptomycin (100 U/ml), and L-glutamine (2 mM)(Gibco Laboratories). The plates were incubated for 4 days at 37 °C and then overlaid with 3 mL of 1% agarose in Medium 199 containing 1% neutral red (Thermo Fisher Scientific). Plaques were counted the following day. The detection limit (one plaque in the well) was 50 PFU/mL, as indicated by a dashed line. Undetectable samples were given a value of 20 PFU for display on the plot.

**BCG immunization**. For BCG immunization, mice were immunized with BCG Pasteur 1173P2 via subcutaneous injection ($2.0 \times 10^5$ CFUs/mouse), and 3 months following BCG immunization, mice were challenged with the aerosolized Mtb K strain. At 4 weeks after the Mtb challenge, mice from each group were euthanized to analyze the bacterial load, histopathology and immune responses. For the coinfection group, mice were i.p. injected with LCMV Arm at 2 weeks post Mtb infection.

**Histopathology**. For histopathological analysis, the right superior lobes of the lungs were preserved in 10% neutral buffered formalin overnight and embedded in paraffin. Then, the lungs were sectioned at 4–5 μm and stained with hematoxylin and eosin (H&E) and Ziehl–Neelsen stain. The severity of inflammation in the lungs was evaluated using ImageJ (National Institutes of Health, USA, v1.53). Briefly, for quantitative histopathological analysis of H&E staining, all slides were scanned using an Olympus BX43 microscope with a 4× objective (Olympus Optical Co., Tokyo, Japan), and a pathologist assessed and quantified histopathology in a blinded manner. Each slide image per mouse was evaluated with the ToupTek Toup viewer program (v4.11, Toup View Co., Zhejiang, China) to quantify the total size of the inflamed lesions in each mouse. The measured area was presented as $mm^2$.

**Cell isolation**. To create a single-cell suspension, each lung was chopped into small pieces and incubated in RPMI supplemented with 1.3 mM EDTA at 37 °C in a shaking incubator for 30 min. After incubation, the small fragments were incubated again in complete medium solution containing 0.1% collagenase type II (Worthington Biochemical, Lakewood, NJ, USA) for 1 h. The single-cell suspensions were then filtered through a 40-μm cell nylon mesh strainer, treated with red blood cell (RBC) lysis buffer (Gibco) for 3 min, and washed twice with RPMI containing 2% FBS. Lymphocytes from LNs were mashed through a 70-μm cell strainer (BD Falcon), and RBC lysis was performed using ACK lysis buffer (Gibco Laboratories).

**Antibodies and flow cytometry analysis**. For phenotypic analysis of lymphocytes, single-cell suspensions were plated at $10^6$ cells/well in 96-well plates (round bottom) in PBS consisting of 2% FBS and stained with fluorochrome-conjugated antibodies for 20 min at 4 °C. Single cell suspensions were strained at a 1:100 dilution; fluorochrome-conjugated antibodies against CD90.2 (Thy1.2; 53-2.1), CD11b (M1/70), CD11c (HL3), Ly6C (AL-21), Ly6G (1A8), NK1.1 (PK136), NKG2D (CX5), and Ki-67 (B56) were obtained from BD Bioscience; antibodies against CD8 (53-6.7), CD19 (eBio1D3), CD44 (IM7), MHCII (M5/114.14.2), Foxp3 (FJK-165), and CXCR3 (CXCR3-173) were obtained from Thermo Fisher; and antibodies against CD4 (RM4-5), CD45.1 (Ly5.1; A20), CD45.2 (Ly5.2; 104), CD25 (PC61), CXCL9 (MIG-2F5.5), and T-bet (4B10) were obtained from Bio-Legend. $D^bMtb32_{93-102}$ monomers conjugated to streptavidin allophycocyanin (Life Technologies, Carlsbad, CA, USA) were used to detect Mtb32-specific CD8+ T cells. For intracellular staining, antibodies against TNF (MP6-XT22) and IFN-γ (XMG1.2) were used in combination with anti-CD4 and anti-CD8 antibodies. The dead cell populations were removed using the Live/Dead Fixable Stain Kit (Thermo Fisher).

To detect intracellular cytokines in T cells from the lung, $10^6$ cells were incubated with each peptide. For Mtb32-specific CD8+ T cell stimulation, $10^6$ cells were incubated with a mixture of Mtb32 peptides (1 μg/mL of each peptide) for 6 h in the presence of GolgiPlug/GolgiStop (BD Bioscience). A total of 33 peptides spanning the mature forms of Mtb32 were synthesized as 17-to-20-mer peptides. For Mtb-specific CD4+ T cell stimulation, the cells were incubated with PPD (Sigma), Ag85B$_{240-254}$ peptide (FQDAYNAAGGHNAVF) (1 μg/mL), and a mixture of ESAT6 peptides (1 μg/mL of each peptide). A total of 14 peptides spanning the mature forms of Mtb ESAT6 were synthesized as 17- and 18-mer peptides. For LCMV-specific T cell stimulation, cells were restimulated with GP$_{33-41}$ and GP$_{276-286}$ peptides (0.2 μg/mL of each peptide) for CD8+ T cell responses or GP$_{66-80}$ peptides (1 μg/mL) for CD4+ T cell responses in the presence of GolgiPlug/GolgiStop (BD Bioscience). All peptides were synthesized by GenScript. To detect CXCL9 in myeloid cells from the lung, $10^6$ cells were incubated with GolgiPlug/GolgiStop (BD Bioscience) for 6 h. Intracellular staining was performed after surface staining using the BD Cytofix/Cytoperm

fixation/permeabilization kit (BD Biosciences) or the Foxp3/transcription factor staining buffer set (Invitrogen) according to the manufacturer's instructions.

**Analysis of in vivo cytokine secretions**. Supernatants of homogenized lung lysates were obtained from each group at the indicated time points. TNF and IL-6 concentrations were measured using a BD cytometric bead array (CBA) mouse inflammation kit (BD Bioscience), and the IFN-α concentrations was measured using a Mouse IFN Alpha ELISA Kit (PBL Assay Science). IFN-γ, IL-10, IL-1α, and IL-1β were measured by a mouse IFN-γ ELISA kit (Thermo Fisher), mouse IL-10 ELISA kit (Thermo Fisher), mouse IL-1α ELISA kit (Ab Frontier), and mouse IL-1β ELISA kit (Ab Frontier). CCL2, CXCL1, and CXCL5 concentrations were measured using a LEGENDplex Multi-Analyte Flow Assay Kit (BioLegend) according to the manufacturer's instructions. The CXCL9 concentration was measured using a Mouse CXCL9 ELISA Kit (R&D Systems), and the CXCL10 concentration was analyzed using a Mouse IP-10 ELISA Kit (Thermo Fisher).

**Single-cell RNA sequencing and single-cell TCR sequencing**. For single-cell RNA sequencing (scRNA-seq) and single-cell TCR sequencing (scTCR-seq), CD45.2+ cells from pooled lung tissues of each group of mice (n = 3) were enriched by magnetic sorting. Single-cell sequencing libraries were prepared by using the Chromium 10X Genomics platform with a Chromium Next GEM Single Cell 5' Library & Gel Bead Kit (10X Genomics) and Chromium Single Cell V(D)J Enrichment Kit for mouse T cells (10X Genomics) according to the manufacturer's instructions. Single-cell suspensions of live sorted pulmonary CD45.2+ cells were diluted in nuclease-free water and then combined with Single-Cell 5' Gel Beads, a master mix, and partitioning oil on Chromium Chip A from the aforementioned kit. RNA transcripts were uniquely barcoded and reverse-transcribed within nanoliter-scale droplets containing uniquely barcoded beads. 10X-barcoded full-length cDNA was then pooled and enriched via PCR. For the TCR-enriched library, the enriched cDNA pool was amplified using primers specific to the T cell receptor (TCR). For 5'Gene Expression (GEX) library, the cDNA pool underwent an end repair process, A-tailing, adaptor ligation, and sample indexing PCR. Both the scRNA-seq and scTCR-seq libraries were sequenced using the HiSeqX platform (Illumina). We used the Seurat R package (R Foundation for Statistical Computing, v3.6.1) to process the data[79,80]. Two different batches were integrated into one dataset using the scMC package (v1.0.0)[81]. We used 40 dimensions from the scMC result for downstream dimension reduction. Cells were clustered using the Louvain algorithm described in Seurat (v4.0.0)[82]. Each cluster was named by its expression of marker genes. The expression of genes in UMAP spaces was visualized using the FeaturePlot function in Seurat. To further analyze the T cells from the experiment, T cell clusters were selected for downstream analysis. We reanalyzed those clusters using the same number of dimensionalities for downstream dimension reductions. For TCR sequencing data, clonotypes assigned to multiple cells were designated expanded clonotypes, and clonotypes assigned to a single cell were designated unique clonotypes. Cells expressing *Ifng* genes over 2.7 were defined as *Ifng*+ cells. A heatmap was drawn using the mean expression of each gene from each sample. The module scores for signature genes from each T cell were calculated from the AddModuleScore function in the Seurat package.

**Cell purification and adoptive cell transfer**. For CD4+ T cell enrichment, CD4+ T cells were isolated from the spleen of wild-type P25 mice by a MagniSort Mouse CD4 T+ T Cell Enrichment Kit (Thermo Fisher). For adoptive transfer of P25 CD4+ T cells (Ly5.1/Ly5.2), the cells were isolated from the spleen of naïve P25 mice after CD4+ T cell enrichment. After isolation, 2×10⁶ purified Ly5.1+CD4+ T cells were transferred into wild-type C57BL/6 mice via the tail vein. Mice were infected with Mtb K 1 day after adoptive transfer.

**In vivo treatment with antibodies and poly I:C**. To block or deplete the indicated targets in vivo, 250 μg of anti-IFNAR1 (MAR1-5A3; Bio X Cell), anti-CD8α (53-6.7; Bio X cell), anti-Gr-1 (RB6-8C5; Bio X cell), anti-IL-10R (1B1.3 A; Bio X cell), and anti-NKG2D antibodies (HMG2D; Bio X cell) or mouse IgG1 isotype control (MOPC-21; Bio X cell), rat IgG2a isotype control (2A3; Bio X cell), rat IgG2b isotype control (LTF-2 l; Bio X cell), rat IgG1 isotype control (HRPN; Bio X cell), and polyclonal Armenian hamster IgG isotype control (Bio X cell) were administered to Mtb K-infected mice on days 13, 15, and 17 by i.p. injection. At 19, 21, 23, and 25 days after LCMV Arm infection, 250 μg of anti-IFNAR1 (MAR1-5A3; Bio X Cell), anti-CD8α (53-6.7; Bio X cell), anti-CD4 (GK1.5; Bio X cell), anti-IL-1R (JAMA-147; Bio X cell), anti-IL-6R (15A7; Bio X cell), anti-TNF (XT3.11; Bio X cell), and anti-Ly6G antibodies (1A8; Bio X cell) or mouse IgG1 isotype control (MOPC-21; Bio X cell), rat IgG2a isotype control (2A3; Bio X cell), rat IgG2b isotype control (LTF-2; Bio X cell), polyclonal Armenian hamster IgG isotype control (Bio X cell), and rat IgG1 isotype control (HRPN; Bio X cell) were administered in vivo. To induce type I IFN production in vivo, poly I:C (200 μg/mouse) (InvivoGen) was intratracheally administered to Mtb K-infected mice at 14, 15 and 16 days.

**RNA isolation and qRT-PCR**. The postcaval lobe of the lung from each group was used for RNA extraction and frozen in RNA*later* (Thermo Fisher) at −80 °C until further use. For RNA isolation, each lobe was chopped into small pieces, or

RAW264.7 cells were harvested and resuspended in 1 mL of TRIzol (Invitrogen). After 10 min of incubation at 4 °C, 200 μL of chloroform (Sigma) was added. The solution was then centrifuged for 20 min at 4 °C, and an equal volume of isopropanol was added to the supernatant to precipitate the RNA. The inverted mixture was centrifuged for 20 min at 4 °C, and the pellet was washed once with 70% ethanol. Finally, the air-dried pellet was dissolved in 80 μL of DEPC-treated water (Invitrogen) and stored at −80 °C until further use.

Total RNA (1 μg) was used to synthesize cDNA via reverse transcription using a Transcriptor First-Strand cDNA Synthesis Kit (Roche) according to the manufacturer's instructions. The expression of individual genes was measured by qPCR on a CFX96 real-time PCR detection system (Bio-Rad) with the following gene-specific forward (F) and reverse (R) primers: *Cxcl9* F: ATG AAG TCC GCT GTT CTT TTC C and R: GTC TCT TAT GTA GTC TTC CTT G; *Cxcl10* F: AGT AAC CCA AGT GCT GCC GTC and R: CTC CAG TTA AGG AGC CCT TTT AG; GAPDH F: AGG TCG GTG TGA ACG GAT TT and R: TGT AGA CCA TGT AGT TGA GG[83]. qPCR reaction conditions were conducted as follows: after initial denaturation of the template for 5 min at 95 °C, 50 thermal cycles of 15 s at 95 °C, 20 s at 60 °C, and 30 s at 72 °C were performed in a final volume of 20 μL using SYBR Green I dye for PCR product detection (Qiagen) according to the manufacturer's instructions. Relative mRNA expression was normalized to *Gapdh* expression and calculated according to the ΔΔCT method.

**Magnetic cell sorting and suppression assays**. Lung cell suspensions were enriched for Ly6G+ populations using two-step magnetic cell sorting. Briefly, lung cells were incubated with CD90.2, CD19, and DX5 microbeads (Miltenyi Biotec) and separated by sequential passaging through an LD MACS column (Miltenyi Biotec). The negative fraction was composed of CD90.2−, CD19-, and DX5− cells. Next, negative fractions were incubated with Ly6G microbeads and separated by sequential passaging through an LS MACS column (Miltenyi Biotec). The positive fraction was mostly composed of Ly6G+ cells.

CD4+ and CD8+ T cells were isolated from the spleens of wild-type C57BL/6 mice. T cells were labeled with 2.5 μM Cell Trace Violet (Life Technologies) for 30 min at 37 °C, washed with RPMI containing 10% FBS, cultured in 96-well plates (flat bottom) in the presence of 2 μg/mL anti-CD28 antibodies (BD Bioscience), and precoated with 5 μg/mL anti-CD3 antibodies (BD Bioscience). In total, 2 × 10⁵ labeled T cells were cocultured with 8 × 10⁵ Ly6G+ cells in RPMI containing 10% FBS. After 72 h, cells were harvested, stained, and analyzed by flow cytometry.

For CD45.2+ cell enrichment, pulmonary cell suspensions were enriched by two-step magnetic cell sorting. Briefly, lung cells were incubated with a CD45.2-FITC antibody (Miltenyi Biotec) and washed out. Then, the cells were incubated with anti-FITC microbeads (Miltenyi Biotec) and separated by sequential passaging through an LS MACS column (Miltenyi Biotec). The positive fraction was mostly composed of CD45.2+ cells.

**Enumeration of CFUs in the Ly6G+ population**. Ly6G+ cells were isolated from the lungs of mice from each group at 3 weeks post Mtb infection using a Myeloid-Derived Suppressor Cell Isolation Kit (Miltenyi Biotec, Bergisch Gladbach, Germany). Isolated Ly6G+ cells from infected and coinfected mice were lysed with 0.05% Triton X-100 (Biosesang, Gyeonggi-go, Republic of Korea), and the lysates were plated onto 7H11 agar to enumerate the bacteria.

**RAW264.7 cell experiment**. RAW264.7 cells, a mouse macrophage-like cell line purchased from the ATCC, were maintained in Dulbecco's modified Eagle's medium (HyClone) supplemented with 10% FBS and 1% penicillin/streptomycin.

For type and type II IFN treatment, 1×10⁵ RAW264.7 cells were plated in 24-well dishes (Costar), and the plates were incubated at 37 °C. After 24 h, 10 or 10² ng/mL mouse IFN-α (PBL Assay Science), mouse IFN-β (PBS Assay Science), and IFN-γ (Peprotech) were added to the appropriate wells. After 24 h, the culture supernatants and RAW264.7 cells were harvested for ELISA and qRT-PCR analysis, respectively.

**Multiplex immunofluorescence staining**. For multiplex immunofluorescence staining, lungs were harvested. Tissue sections (4.5 μm) were cut from formalin-fixed paraffin-embedded (FFPE) blocks and dried for at least 1 h in a 60 °C dry incubator. The sections were dewaxed using xylene, followed by multiplex immunofluorescence staining with a Leica Bond Rx Automated Stainer (Leica Biosystems, Newcastle, UK).

Briefly, the slides were baked for 30 min and dewaxed with Leica Bond Dewax solution (Leica Biosystems, AR9222), followed by Ag retrieval with Bond Epitope Retrieval 2 (Leica Biosystems, AR9640) in a pH 9.0 solution for 30 min. Then, the slides were stained for CD31 (dilution 1:500, polyclonal, Abcam, ab124432, Opal 520), CXCL9 (dilution 1:100, polyclonal, Abcam, ab202961, Opal 570), CD11b (dilution 1:4000, EPR1344, Abcam, ab133357, Opal 690), and PDPN (dilution 1:2000, polyclonal, Abcam, ab109059, Opal 620). Nuclei were subsequently visualized with DAPI staining (Akoya Biosciences, FP1490), and the sections were coverslipped using HIGHEDF IHC fluormount (Enzo Life Science, ADI-950-260-0025). The slides were scanned using the Vectra Polaris Automated Quantitative Pathology Imaging System (Akoya Bioscience), and images were analyzed using inform 2.4 software and TIBCO Spotfire (Akoya Bioscience). To count CXCL9+

cells, we randomly selected noninflammatory and inflammatory regions and then counted DAPI and CXCL9-copositive cells.

**Quantification and statistics**. All stained samples were assessed on a FACS Canto II instrument (BD Bioscience) or CytoFLEX LX (Beckman Coulter) and analyzed using FlowJo software (Tree star, v10.5.3).

Statistical analysis was performed using Prism software version 7.0 (GraphPad, v3.6.1). Two-tailed unpaired Student's t tests were performed to determine differences between two groups. Comparisons between multiple groups were performed using one-way ANOVA with a *post hoc* Tukey test or two-way ANOVA with a *post hoc* Tukey test. Kaplan-Meier survival curves were analyzed using the Mantel-Cox log-rank test with a 95% CI. The statistical details, including the statistical test, exact value of n, precision measure and statistical significance threshold, are reported in the figures and figure legends.

**Reporting summary**. Further information on research design is available in the Nature Research Reporting Summary linked to this article.

## Data availability
The scRNA-seq and scTCR seq data generated in this study have been deposited in the Gene Expression Omnibus database under primary accession code GSE167650. All other data are available in the article and its Supplementary files or from the corresponding author upon reasonable request. Source data are provided with this paper.

## Code availability
No custom code or algorithms were used in this study.

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

## Acknowledgements

This study was supported by National Research Foundation of Korea (NRF) grants funded by the Korean government (MSIT) (2019R1A2C2003204 to S.J.S.; 2017R1A5A1014560, 2019M3A9B6065221 to S.H.). This study was also supported by the Korean Health Technology R&D Project (HV20C0144 to S.J.S. and S.H.) through the Korean Health Industry Development Institute (KHIDI) funded by the Ministry of Health & Welfare. The funders had no role in study design, data collection and analysis, decision to publish, or preparation of the manuscript.

## Author contributions

Conceptualization; S.H. and S.J.S.; Experimentation; T.G.K., K.W.K. and M.J.K.; Data analysis; T.G.K., K.W.K. and I.L.; Writing and editing the manuscript, T.G.K., K.W.K., S.H. and S.J.S.; Funding acquisition; S.H. and S.J.S.; Supervision, S.H. and S.J.S.

## Competing interests

The authors declare no competing interests.
