## [Peer Review File · Nature Communications]

Viral coinfection promotes tuberculosis immunopathogenesis by type I IFN signaling-dependent impediment of Th1 pulmonary influxREVIEWER COMMENTS

Reviewer #1 (Remarks to the Author):

In this article the authors investigate the impact of viral infection on the development of immunity in a mouse model of aerosol infection with Mtb. They found that infection with LCMV at 14 days post Mtb challenge resulted in increased bacterial accumulation, pyogranulomatous inflammation and greater overall disruption of the lung tissue (fig 1). Temporal examination of the lung tissue post LCMV delivery showed increased neutrophil accumulation, as well as increased accumulation of TNF, IL-6 (transitory), CCL2, CXCL1 and CXCL5. There was an increase in type I IFN - which was followed up. There was a lower frequency of antigen responsive IFN/TNF producing T cells in the LCMV coinfecting mouse lungs. Inhibition of the type I receptor proximal to the delivery of the LCMV abrogated the effect on pathology, bacterial number increase and frequency of cytokine producing cells (ab inhibition and KO mice both used). Analysis of CXCL9 in non-inflammatory regions of the lung was reduced by co-infection with LCMV and this effect was abrogated by blocking of type I IFN Receptor. Using TCRTg T cells the impact of LCMV infection at day 14 on accumulation of antigen specific T cells in the lung at day 21 was addressed. It was found that type I IFN induced by the infection reduced accumulation of antigen specific T cells as well as the number that had proliferated, become activated and could produce cytokine. There was also an effect on CXCL9 and 10 protein production. There was a correlation between the type of T cells and CFU. There was also a correlation between CXCL9 and CXCL10 protein production and T cells in the lung. Finally the ability of type I IFNs to inhibit type II IFN induction of chemokines was determined in an in vitro system. All together the data support a working model wherein type I IFN induced by viral infection can act to limit the induction CXCL9 in lung cells and thereby compromise accumulation of antigen specific cytokine producing T cells into the lungs in a timely manner thereby allowing Mtb to grow more quickly resulting in greater accumulation of inflammatory damage. The data fails to definitively prove the causal connection between the LCMV effects on CXCL9 and 10 and the inflammatory and bacterial growth consequences. Overall the experiments appear to be excellently performed and the materials and methods are thorough.

1. Due to the lack of the causal connection data the conclusions in the latter part of the abstract are overstated. This can be addressed by more cautious language but it does reduce the significance of the paper.
2. There is a strong dependence upon changes in percentage of cell types rather than absolute numbers of cells. Frequencies in one population can alter frequencies of other populations in a manner and this can limit interpretation of frequency data. Sometimes the authors discuss cell numbers and express data in terms of more and less cells - this is not appropriate if only frequencies are reported.
3. The bacterial dose of 300 is quite large but this may have been required to see the effect noted. At a dose of 300 any minor deficiency in time to accumulation of T cells will result in the kind of pathology shown in this paper.
4. Fig 1 panels g, h, and i likely reflect slight differences in doubling rate in the lung prior to immune cell accumulation rather than any 'lab adaptation'. If you keep these panels please do not use the phrase lab-adapted as it is undefined and reduces clarity of interpretation.
5. The BCG data seemed to add a level of complexity and was interpreted to mean that there were lung resident T cells. This was an over interpretation and I don't think these data add to the message. It is again likely that the increased frequency of circulating antigen specific T cells were able to accumulate quickly enough to stop bacterial outgrowth and thereby limit the consequences of the type I inhibition of CXCL9 expression.
6. Fig 3 e data needs the MtbK alone data to put the data in context.
7. As a reader I was overwhelmed by the data sets - particularly Fig 4. It might be prudent to limit

the data sets to those that support the conclusions. The PCR data for CXCL 9 and the histology data provide that information without the complexity and distraction of fig 4.

Overall, a well performed and clearly reported set of experiments. There was too much extraneous data which clouded the message of the paper. There was also overstatement of the conclusions based on two sets of correlative data rat5her than one causla data set.

Reviewer #2 (Remarks to the Author):

In this manuscript entitled "Viral coinfection promotes tuberculosis immunopathogenesis by type I IFN signaling-dependent impediment of Th1 pulmonary influx", the authors investigate how coinfection with LCMV Armstrong perturbs the Th1 response during Mtb infection, and ultimately culminates in excessive inflammation and immunopathology in the lungs, as well as a failure to control bacterial growth. The study is quite comprehensive, and the authors use multiple experimental approaches to show that LCMV-mediated induction of type I IFNs (during a certain time window) is responsible for restricting IFN- γ -producing CD4+ and CD8+ T cell responses in the lung, and is also the driving force behind the pulmonary pathology that occurs following coinfection with LCMV and Mtb. While there have been several previous reports demonstrating a detrimental role for type I IFNs in Mtb pathology, the authors' extensive investigation into multiple immune cell subsets and parameters, identification of when type IFN signaling suppresses T-cell-mediated immunity, and mechanistic insight into how type I IFN signaling might function to limit T cell migration to the lung (via suppressing Cxcl9 and Cxcl10 production) are all important contributions to the field. The conclusions from this paper could however be further strengthened upon inclusion of some additional experimental data and controls.

Specific comments:

The anti-IFNAR blocking experiments and complementary use of *Ifnar1*^{-/-} mice do clearly show that much of the pathology caused by coinfection with LCMV Arm and Mtb is due to type I IFN signaling. However, control over bacterial outgrowth and T cell functionality are not fully restored to that of control mice upon either α -IFNAR blocking or in *Ifnar1*^{-/-} mice, suggesting that some IFN-independent mechanisms spurred by LCMV infection may also contribute, and this should be noted. This is further supported by the authors finding that administration of poly I:C does not fully phenocopy the effects of LCMV and Mtb coinfection, although there does appear to be some trends towards diminished IFN γ and TNF α production by Mtb-specific T cells, as well as impaired control over bacteria replication. The experiments with Poly I:C administration does bring to question however how effective type I IFN production was induced here, as the concentration of IFN- α appears relatively low compared to LCMV Arm-infection, although it appears that IFN levels were only measured at 3 and 6 hours post Poly I:C administration. Do IFN levels spike by 24 or 48 hours post Poly IC administration to levels similar to those found during LCMV infection? The argument that LCMV Arm spurs high and sustained IFN production compared to Poly I:C (line 287), and that this is likely the reason behind the augmented lung pathology that occurs more selectively during co-infection versus Mtb + Poly I:C treatment is not fully supported by the data. IFN- α levels would have to be shown at later time points post Poly I:C treatment. Thus, the authors may have to refine these conclusions a little bit unless this data can be shown. Additionally, I do wonder if additional Poly I:C injections (perhaps on days 14, 15, and 16) post Mtb infection would better recapitulate the results of Mtb and LCMV coinfection, and strengthen the authors conclusions that type I IFNs are really the culprit in driving lung immunopathology during Mtb infection, although this could definitely be considered busy work.

Figure 4 e-f. Although the heatmap in 4e does appear to shows that Cxcl9 and Cxcl10 expression are decreased in the macrophage 2 cluster during coinfection, and that expression of these chemokines can partly be rescued by α -IFNAR treatment, the violin plots in 4f do not look all that

convincing. Is there another way to display this data? Further, the relevance of this macrophage cluster is not quite clear. Is this subset the predominant source of Cxcl9 and Cxcl10 during Mtb infection? Would it be helpful to compare Cxcl9 and Cxcl10 expression among the macrophage, monocyte, and neutrophil clusters during Mtb infection alone? Is it possible to assess whether the macrophage 2 cluster has increased IFN γ -STAT1 signaling compared to the other myeloid clusters, which may contribute to its high expression of Cxcl9?

Figure 4h. Do the effector CD4 T cell clusters express Cxcr3 and Cx3cr1? Is there a difference in expression of these Th1 molecules between experimental conditions?

Figure 6: The generation and use of P25 transgenic CD4⁺ T cells is a great approach to study how co-infection and type I IFN signaling impacts CD4 T cell differentiation, migration and function. However, expression of CD44, ki67 and cytokines IFN- γ and TNF- α are the only parameters assessed here, and additional characterization of these Mtb-specific CD4 T cells would be helpful. Is there a difference in expression of CXCR3, CX3CR1, T-bet etc.? Given that a central component of the study is focused on CXCR3-ligands Cxcl9 and Cxcl10, expression of CXCR3 on P25 cells should be shown for all experimental groups, and a comparison of other Th1-associated markers would be helpful. If the authors do not have this data on hand, they should at least show the expression of these markers in their scRNA seq data for all experimental groups. Additionally, do the authors have data showing the kinetics of P25 cells in the lungs over time? Is there a major influx/accumulation of P25 cells in the lungs between 2-3 weeks post infection, which would fit with their overall model that coinfection with LCMV starting at 3 weeks post Mtb infection has no impact on pathology, likely due to establishment of robust Th1 responses by that time, whereas mice are still susceptible at 2 weeks post-infection which the authors suggest is due to insufficient Th1 responses at that time.

As Cxcl10 is a known type I IFN-stimulated gene, it is somewhat surprising that Cxcl10 levels increase in the lungs after IFNAR blockade. Can the authors comment on this more in the discussion section? Is it thought that the increase in IFN- γ production upon IFNAR blockade is sufficient to amplify Cxcl10 production even in the absence of IFNAR signaling?

Reviewer #3 (Remarks to the Author):

This report investigates Mycobacterium tuberculosis and acute Lymphocytic choriomeningitis virus co-infection in mice. Mtb infection is a significant global public health concern and developing a understanding the impact of co-infections on the outcome and severity of disease is important for both deciphering the mechanisms of pathogenesis as well as for devising strategies to alleviate the infection. The authors tackle this by establishing Mtb infection in mice and subsequently eliciting an acute viral infection with LCMV. Co-infection with LCMV during a precise window of time, corresponding to the immune recruitment phase during Mtb, increases the severity of the Mtb infection when certain clinical isolates but not laboratory isolates are used. Further analyses of clinical Mtb isolate and acute LCMV co-infection demonstrates that this scenario results in increased neutrophilia and decreased Mtb-specific T cell responses. These alterations are somewhat reversible by blocking type-I IFN. The roles of type-I IFN in enhancing pathogenesis are further supported by experiments in which poly I:C was administered. One of the major changes that occurred during acute LCMV infection was to the inflammatory infiltrate and the expression of certain chemokines including CXCL9 and CXCL10, which impact the recruitment of Mtb specific T cells to the lung. Overall the studies highlight a potential role of type-I IFNs in enhancing Mtb pathogenesis by altering the inflammatory infiltrate, T cell response, and composition of the cytokine milieu. Although the report contains a wealth of data and presents some interesting

results my level of enthusiasm for the report is mixed. Although model systems can be highly informative, I find this strategy odd. The disease enhancement phenomena is only observed with certain isolates of Mtb and it is not clear why at least some amount of augmentation is not observed with the non-clinical strains? It is interesting, but arguably under-developed, why the enhancement is so dependent on the timing of LCMV infection. Notwithstanding the aforementioned concerns, given the data presented it is also likely that factors other than type-I interferon levels also impact the outcome of Mtb infection.

Additional points that the authors should consider are listed below:

1. The authors conclude that increased IFN-I levels alter the innate and adaptive response to Mtb. What remains unclear is whether IFN-I is acting directly or indirectly on the responses. For example, are T cell responses diminished due to IFN-I induced changes to myeloid cells or does the direct action of IFN-I on the priming T cells also impact the response etc?
2. As the authors acknowledge CXCL9 and CXCL10 levels were partially restored by IFNAR blockade (lines 317-320). This is an important point. Blocking type-I IFN in the co-infected animals clearly does not fully rescue the responses and revert them to those observed with Mtb alone. This is suggested by fig 4e, but also 4f, where although the anti-IFNAR treatment does change the response, it appears different to that elicited by Mtb only and the Mtb with LCMV coinfection. Similarly, in figure 5 the responses are also not fully rescued. While I agree that there is a likely temporal contribution of IFN-I there are very likely other factors involved. Thus, important gaps remain in our understanding of this phenomena which are not resolved by the current report.
3. In figure 2E IFN-gamma levels are analyzed at 21 days and are less than 2-fold different in the co-infection group. Is this difference sufficient to account for the alterations in bacterial clearance etc, and is this a transient or more substantial change? What is the profile if levels are checked over time?
4. In figure 2F, the source of the specimens shown in the upper and lower panels should be clearly indicated.

Responses to reviewer's comments

Reviewer #1 (Remarks to the Author):

In this article the authors investigate the impact of viral infection on the development of immunity in a mouse model of aerosol infection with Mtb. They found that infection with LCMV at 14 days post Mtb challenge resulted in increased bacterial accumulation, pyogranulomatous inflammation, and greater overall disruption of the lung tissue (fig 1). Temporal examination of the lung tissue post LCMV delivery showed increased neutrophil accumulation, as well as increased accumulation of TNF, IL-6 (transitory), CCL2, CXCL1 and CXCL5. There was an increase in type I IFN - which was followed up. There was a lower frequency of antigen responsive IFN/TNF producing T cells in the LCMV coinfecting mouse lungs. Inhibition of the type I receptor proximal to the delivery of the LCMV abrogated the effect on pathology, bacterial number increase and frequency of cytokine producing cells (ab inhibition and KO mice both used). Analysis of CXCL9 in non-inflammatory regions of the lung was reduced by co-infection with LCMV and this effect was abrogated by blocking of type I IFN Receptor. Using TCR Tg T cells the impact of LCMV infection at day 14 on accumulation of antigen specific T cells in the lung at day 21 was addressed. It was found that type I IFN induced by the infection reduced accumulation of antigen specific T cells as well as the number that had proliferated, become activated and could produce cytokine. There was also an effect on CXCL9 and 10 protein production. There was a correlation between the type of T cells and CFU. There was also a correlation between CXCL9 and CXCL10 protein production and T cells in the lung. Finally the ability of type I IFNs to inhibit type II IFN induction of chemokines was determined in an *in vitro* system. All together the data support a working model wherein type I IFN induced by viral infection can act to limit the induction CXCL9 in lung cells and thereby compromise accumulation of antigen specific cytokine producing T cells into the lungs in a timely manner thereby allowing Mtb to grow more quickly resulting in greater accumulation of inflammatory damage. The data fails to definitively prove the causal connection between the LCMV effects on CXCL9 and 10 and the inflammatory and bacterial growth consequences. Overall, the experiments appear to be excellently performed and the

materials and methods are thorough.

1. Due to the lack of the causal connection data the conclusions in the latter part of the abstract are overstated. This can be addressed by more cautious language but it does reduce the significance of the paper.

Response: We deeply appreciate the reviewer's valuable comments. The reviewer pointed out that in the latter part of the abstract, the results clearly showing the causal connection from coinfection to exacerbation of TB pathogenesis are somewhat insufficient. Our data evidently showed that TB-virus coinfection induced more type I IFN production, inhibited pulmonary CXCL9 expression and the Mtb-specific Th1 response, and exacerbated pulmonary pathogenesis. Although we have already demonstrated *in vitro* that IFN- γ -induced CXCL9 expression occurs in the presence of type I IFN, we could not directly demonstrate *in vivo* the causal connection that type I IFN production induced upon viral infection hinders CXCL9 production in the lung, which subsequently limits the migration of Mtb-specific Th1 cells into the lung. Instead, we performed additional experiments using the repeated intratracheal administration of Poly I:C (**Supplementary Fig. 7**), which induced a high level of increased and sustained type I IFNs in lungs similar to that induced by viral coinfection (**Supplementary Fig. 8**). We obtained similar results including weight loss of mice, a high number of bacterial loads with hyperinflammation, the delayed pulmonary Th1 response (**Supplementary Fig. 7**), and a low level of CXCL9-expressing myeloid cells (**Supplementary Fig 13c and 13d**). These impaired Mtb-specific protective responses were rescued when treating mice with type I IFN receptor blocking antibody like virus coinfection (**Supplementary Fig. 7f**). Thus, we believe that exacerbated Mtb immunopathology by virus coinfection was principally evoked in a sustainability and degree of type I IFN-dependent manner, rather than another effect of virus.

Also, it is well known that CXCL9/10 is a critical factor for the migration of effector T cells to infected or inflamed sites (CXCR3 ligand: redundant, collaborative, and antagonistic function; Immunol. Cell Biol; 89:207-215; 2011) and that rapid migration of Mtb-specific T cells to the pulmonary site is mostly important for protection against Mtb (Control of Mycobacterium tuberculosis infection by a subset of lung parenchyma-homing CD4 T cells; J. Immunol; 192:2965-2969; 2014). Therefore, based on our *in vitro* data showing type I IFN-mediated inhibition of CXCL9 production and these references, it is conceivable that curtailed expression

of CXCL9 due to viral coinfection-induced type I IFN reduces the migration of Mtb-specific T cells to pulmonary lesions, which would result in increased bacterial growth *in vivo*.

To directly demonstrate this causal connection *in vivo*, it is thought to be essential to use conditional knockout mice, such as $\text{Ifnar1}^{\text{fl/fl}} \times \text{CD4}^{\text{Cre-ERT2}}$ and $\text{Ifnar1}^{\text{fl/fl}} \times \text{LysM}^{\text{Cre-ERT2}}$ mice, which can spatiotemporally regulate the expression of type I IFN receptors on T cells and myeloid cells, respectively, because the *Ifnar1* gene should be knocked out at the time of virus infection but not during Mtb infection. We think that this approach takes too much time and effort to technically perform these experiments using knockout mice.

Therefore, according to the reviewer's suggestion, we have rewritten the conclusion section in the abstract to avoid overstatement, as follows.

“Single-cell sequencing, tissue immunofluorescence staining, and adoptive transfer experiments indicated that viral infection-induced type I IFN signaling could inhibit CXCL9/10 production in myeloid cells, ultimately impairing pulmonary migration of Mtb-specific CD4⁺ T cells. Thus, our study suggests that augmented and sustained type I IFNs by virus coinfection prior to the pulmonary localization of Mtb-specific Th1 cells exacerbates TB immunopathogenesis by impeding the Mtb-specific Th1 cell influx. Our study highlights another novel negative role of viral coinfection-induced type I IFN responses in delaying Mtb-specific Th1 responses in the lung.” [Lines 11-18]

2. There is a strong dependence upon changes in percentage of cell types rather than absolute numbers of cells. Frequencies in one population can alter frequencies of other populations in a manner and this can limit interpretation of frequency data. Sometimes the authors discuss cell numbers and express data in terms of more and less cells - this is not appropriate if only frequencies are reported.

Response: We agree with the reviewer's opinion. As the reviewer suggested, we provided new data showing the absolute numbers of cells. Similar to the frequency of CD11b⁺Ly6G⁺ cells, the absolute number of CD11b⁺Ly6G⁺ cells was increased at 7 days post LCMV Arm infection in the coinfecting group (**Fig. 2a**). Similar to the change in the frequency of Mtb-specific CD4⁺ and CD8⁺ T cells among each subset of T cells in the lungs, the absolute numbers of IFN- γ -

producing ESAT6-specific CD4⁺ T cells and Mtb32-specific CD8⁺ T cells were decreased in the coinfecting group compared to those in the Mtb only-infected group, but the numbers were rescued by type I IFN receptor blockade (**Fig. 3e-f**). We also included the absolute numbers of CXCL9⁺ cells of each myeloid subset in the lung (**Supplementary Fig. 10c, Supplementary Fig. 13b and 13d**) and absolute numbers of P25 cells in the dLN and/or lung (**Fig. 7b and Supplementary Fig. 12a**). In addition, we also calculated the correlation between the absolute numbers of P25 or IFN- γ ⁺ P25 cells and CFUs in the lung (**Supplementary Fig. 12d**). Accordingly, we have added the following text to the revised manuscript.

*“We found a striking and gradual increase in the frequency and **number** of neutrophils at 7 days post LCMV Arm infection (Fig. 2a),” [Lines 129-131]*

*“Furthermore, coinfecting mice with curtailed Mtb-specific Th1 immunity regained CD4⁺ and CD8⁺ T cell functionality both without and with ex vivo restimulation, according to both their frequency and **absolute number** (Fig. 3e and 3f)” [Lines 213-216]*

*“We also found that the frequency and **number** of CD11b⁺F4/80⁺CXCL9⁺ cells significantly decreased in the coinfecting group but this decrease was rescued by type I IFN receptor blockade (Supplementary Fig. 10b and 10c)” [Lines 333-336]*

*“In addition, we further analyzed the kinetics of P25 cells in the lungs. The frequency and **number** of P25 cells at 3 weeks post Mtb infection were higher than those at 2 weeks post Mtb infection, which suggested that a major influx and accumulation of P25 cells occurred between 2 and 3 weeks after Mtb infection (Supplementary Fig. 12a)” [Lines 401-405]*

*“Additionally, the lung bacterial loads were inversely correlated with the frequency and **number** of P25 cells or IFN- γ ⁺ P25 cells in the lung, which suggests that Mtb-specific T cells and their functions in the lungs play a central role in host protection from pulmonary pathology induced by viral coinfection (Fig. 6e and Supplementary Fig. 12d)” [Lines 422-426]*

*“We found that the frequency and **number** of P25 cells in the lungs significantly decreased in coinfecting mice compared to those in Mtb alone-infected mice, whereas those in the dLNs were not different in these two groups of mice (Fig. 7a and 7b)” [Lines 434-437]*

3. The bacterial dose of 300 is quite large but this may have been required to see the effect noted. At a dose of 300 any minor deficiency in time to accumulation of T cells will result in the kind of pathology shown in this paper.

Response: To address the reviewer's question, we added new supplementary data showing the degree of TB immunopathology from infections with lower doses of Mtb K or LCMV Arm (**Supplementary Fig. 1g-i**). Similar pathologies were observed even at lower doses of Mtb K infection (bacterial doses: 130 and 230 CFU). When we also investigated the effect of infection with a half dose of LCMV Arm (1×10^5 PFU), the mice exhibited a similar range of pulmonary pathologies (**Supplementary Fig. 1g-i**). Overall, coinfection-mediated pulmonary pathology was still observed even with the infections with lower doses of Mtb K or LCMV Arm. We added the following text to the revised manuscript.

"We further analyzed the dose-dependent effect of bacteria or viruses on pulmonary pathology. When mice were infected with a lower dose of Mtb or LCMV Arm (Supplementary Fig. 1g-i), they also exhibited a similar range of pulmonary pathology." [**Lines 114-117**]

4. Fig 1 panels g, h, and i likely reflect slight differences in doubling rate in the lung prior to immune cell accumulation rather than any 'lab adaptation'. If you keep these panels please do not use the phrase lab-adapted as it is undefined and reduces clarity of interpretation.

Response: We completely agree with the reviewer's opinion, and thus, we deleted the phrase "lab-adapted" in the revised manuscript. We modified the text in the revised manuscript as follows.

"Next, we infected mice with diverse Mtb strains (H37Ra, H37Rv, Erdman, HN878, K, and M2) to investigate a possible association between aggravated immunopathology and different Mtb strains. Mice infected with HN878, K, or M2 showed exacerbated pathology along with excessive bacterial growth in the lung, whereas those infected with H37Ra, H37Rv, or Erdman showed less severe pathological profiles (Fig. 1h-j) and bacterial loads comparable to those with Mtb infection alone. Thus, pathology exacerbation was dependent on the Mtb strain." [**Lines 119-124**]

5. The BCG data seemed to add a level of complexity and was interpreted to mean that there were lung resident T cells. This was an over interpretation and I don't think these data add to the message. It is again likely that the increased frequency of circulating antigen specific T cells were able to accumulate quickly enough to stop bacterial outgrowth and thereby limit the consequences of the type I inhibition of CXCL9 expression.

Response: We appreciate the valuable comment and agree with your opinion. However, in contrast to the reviewer's comment, our manuscript did not state that lung resident T cell numbers increased after BCG vaccination. As the reviewer pointed out, it is unknown whether BCG vaccination contributed to the generation of Mtb-specific lung resident T cells and/or the increase in the number of circulating Mtb-specific T cells. However, based on our data, BCG vaccination contributed to an increase in the Mtb-specific Th1 response in the lung, the alleviation of severe pulmonary pathology, and the control of bacterial outgrowth after Mtb and virus coinfection, as shown in our previous data (**Supplementary Fig. 3b-g**).

In the revised manuscript, we examined whether BCG vaccination indeed can increase the number of Mtb-specific Th1 cells in the lung post Mtb infection alone without LCMV infection. In a newly conducted experiment, we compared the Mtb-specific Th1 response in BCG-vaccinated and unvaccinated mice at 14 days post Mtb infection. We observed that both the frequency and number of CD4⁺ T cells producing IFN- γ upon restimulation with PPD significantly increased in the lungs of BCG-vaccinated mice but not in those of unvaccinated mice (**Supplementary Fig. 3a**). More interestingly, BCG vaccination also contributed to the increased production of CXCL9 from myeloid populations, including CD11b⁺Ly6G⁺, CD11b⁺Ly6C⁺, and CD11b⁺F4/80⁺ cells, post Mtb infection (**Supplementary Fig. 13a and 13b**). Collectively, these data together with our previous data suggest that accelerated generation of pulmonary Mtb-specific Th1 cells post Mtb infection in BCG-vaccinated mice compared to that in unvaccinated mice could contribute to IFN- γ -induced CXCL9 production in the lung, which subsequently leads to a rapid accumulation of Mtb-specific Th1 cells that are sufficient to control bacterial outgrowth and pulmonary pathogenesis.

Accordingly, we modified the text in the revised manuscript as follows.

“At 14 days post Mtb infection, Mtb-infected mice without BCG vaccination barely elicited IFN- γ responses in the lung, whereas Mtb-infected mice with BCG vaccination showed significantly increased frequencies and numbers of Mtb-specific CD4⁺ T cells that were able to produce IFN- γ upon restimulation with purified protein derivative (PPD) (Supplementary Fig. 3a). Despite viral coinfection, the BCG-vaccinated group displayed less severe pulmonary pathology, fewer granuloma lesions (Supplementary Fig. 3b and 3c), lower bacterial loads in the lung (Supplementary Fig. 3d) and less neutrophil infiltration (Supplementary Fig. 3e) than the unvaccinated group. Additionally, cells from the BCG-vaccinated and coinfecting group secreted significantly more IFN- γ (Supplementary Fig. 3f and 3g) upon restimulation with PPD at 21 days post Mtb infection than cells from the unvaccinated and coinfecting groups, suggesting that the BCG-primed generation of Mtb-specific T cells in the lung might ameliorate the exacerbated pulmonary pathology induced by viral coinfection.” [Line 180-191]

“In addition, the BCG-vaccinated group, which showed ameliorated exacerbation of pulmonary pathology induced by viral coinfection, already exhibited enhanced expression of CXCL9 at the time of LCMV Arm coinfection (14 days post Mtb infection) (Supplementary Fig. 13a and 13b). These data together with the result showing a higher Mtb-specific Th1 immune response in the lungs of the BCG-vaccinated mice (Supplementary Fig. 3a-g) suggest that accelerated generation of pulmonary Mtb-specific Th1 cells post Mtb infection in BCG-vaccinated mice, compared to that in unvaccinated mice, could contribute to IFN- γ -induced CXCL9 production in the lung, which subsequently leads to rapid accumulation of Mtb-specific Th1 cells that are sufficient to control bacterial outgrowth and pulmonary pathogenesis.” [Lines 466-475]

6. Fig 3e data needs the Mtb K alone data to put the data in context.

Response: We appreciate your comment. We agree with the reviewer’s suggestion. As the reviewer recommended, we added data for the Mtb K alone condition in **Fig. 3e and 3f**.

7. As a reader I was overwhelmed by the data sets - particularly Fig 4. It might be prudent to limit the data sets to those that support the conclusions. The PCR data for CXCL 9 and the histology data provide that information without the complexity and distraction of fig 4.

Response: We agree with the reviewer's opinion. As the reviewer suggested, we modified **Fig. 4**. Since the data (**Fig. 4e and 4f** in the original manuscript) showing the difference in CXCL9 and CXCL10 expression in myeloid populations among the Mtb K alone, coinfection, and α IFNARI plus coinfection groups were essential to the design of the following experiments, including histology and qRT-PCR, we think that it is inappropriate to move or delete the figures. Instead, due to the complexity of Fig. 4, we moved the two figures (**Fig. 4c and Fig. 4h** in the original manuscript) into **Supplementary Fig. 9a and Supplementary Fig. 11a** in the revised manuscript. We think that these revisions made **Fig. 4** much simpler.

Reviewer #2 (Remarks to the Author):

In this manuscript entitled “Viral coinfection promotes tuberculosis immunopathogenesis by type I IFN signaling-dependent impediment of Th1 pulmonary influx”, the authors investigate how coinfection with LCMV Armstrong perturbs the Th1 response during Mtb infection, and ultimately culminates in excessive inflammation and immunopathology in the lungs, as well as a failure to control bacterial growth. The study is quite comprehensive, and the authors use multiple experimental approaches to show that LCMV-mediated induction of type I IFNs (during a certain time window) is responsible for restricting IFN- γ -producing CD4⁺ and CD8⁺ T cell responses in the lung, and is also the driving force behind the pulmonary pathology that occurs following coinfection with LCMV and Mtb. While there have been several previous reports demonstrating a detrimental role for type I IFNs in Mtb pathology, the authors’ extensive investigation into multiple immune cell subsets and parameters, identification of when type IFN signaling suppresses T-cell-mediated immunity, and mechanistic insight into how type I IFN signaling might function to limit T cell migration to the lung (via suppressing Cxcl9 and Cxcl10 production) are all important contributions to the field. The conclusions from this paper could however be further strengthened upon inclusion of some additional experimental data and controls.

Specific comments:

1. The anti-IFNAR blocking experiments and complementary use of *Ifnar1*^{-/-} mice do clearly show that much of the pathology caused by coinfection with LCMV Arm and Mtb is due to type I IFN signaling. However, control over bacterial outgrowth and T cell functionality are not fully restored to that of control mice upon either a-IFNAR blocking or in *Ifnar1*^{-/-} mice, suggesting that some IFN-independent mechanisms spurred by LCMV infection may also contribute, and this should be noted. This is further supported by the authors finding that administration of poly I:C does not fully phenocopy the effects of LCMV and Mtb coinfection, although there does appear to be some trends towards diminished IFN- γ and TNF- α production by Mtb-specific T cells, as well as impaired control over bacteria replication.

Response: We deeply appreciate the reviewer’s comments. As the reviewer pointed out, Mtb-specific T cells are not fully rescued by type I IFN receptor blockade. We also agree that some

type I IFN-independent mechanism stimulated by LCMV infection may also contribute to the exacerbation of pulmonary pathology. As shown in **Supplementary Fig. 5**, we conducted several experiments by using diverse monoclonal antibodies to validate the effect of some immunological factors, which were enriched due to LCMV infection in our coinfection model. Unfortunately, no other factors contributing to exacerbation of pathology other than type I IFN were found, which suggests that pulmonary pathology in coinfecting group could be derived from the sum of many factors, and the main effector among them seems to be type I IFNs. Since the risk of TB-associated mortality due to coinfection with other infectious agents has been increasingly reported, further research should be conducted with diverse mouse coinfection models. We have included the reviewer's concern as a limitation of our study in the Discussion section of the revised manuscript as follows.

“Furthermore, by using IFNAR-1 blockade and complementary application of type I IFN receptor knockout mice, we clearly show that much of the exacerbated pathology caused by coinfection is due to type I IFN signaling. However, Mtb-specific T cell responses or expression of CXCL9 and CXCL10 was not fully restored by IFNAR-1 blockade, which suggests that some type I IFN-independent mechanisms stimulated by LCMV infection may also contribute to exacerbated pulmonary pathology in coinfecting mice. Since the risk of TB-associated mortality due to coinfection with other infectious agents has been increasingly reported, further research should be conducted with diverse mouse coinfection models.” [Lines 531-538]

2. The experiment with Poly I:C administration does bring to question, however, how effective type I IFN production was induced here, as the concentration of IFN- α appears relatively low compared to LCMV Arm-infection, although it appears that IFN levels were only measured at 3 and 6 hours post Poly I:C administration. Do IFN levels spike by 24 or 48 hours post Poly IC administration to levels similar to those found during LCMV infection? The argument that LCMV Arm spurs high and sustained IFN production compared to Poly I:C (line 287), and that this is likely the reason behind the augmented lung pathology that occurs more selectively during co-infection versus Mtb + Poly I:C treatment is not fully supported by the data. IFN- α levels would have to be shown at later time points post Poly I:C treatment. Thus, the authors may have to refine these conclusions a little bit unless this data can be shown. Additionally, I do wonder if additional Poly I:C injections (perhaps on days 14, 15, and 16) post Mtb infection

would better recapitulate the results of Mtb and LCMV coinfection, and strengthen the authors conclusions that type I IFNs are really the culprit in driving lung immunopathology during Mtb infection, although this could definitely be considered busy work.

Response: We appreciate the reviewer's valuable suggestions. As the reviewer suggested, we designed a new experiment in which Mtb K-infected mice were injected intratracheally with poly I:C three times at 14, 15, and 16 days post Mtb infection (**Supplementary Fig. 7a**). As a result, the pulmonary pathology became evidently severe after multiple poly I:C injections and was dramatically alleviated by type I IFN receptor blockade (**Supplementary Fig. 7b and c**). Since poly I:C-injected mice looked very sick, we sacrificed them at an earlier time point, 26 days post Mtb infection, than at the time point used in other experiments, 31 days post Mtb infection. The body weight and bacterial load data also supported our observation (**Supplementary Fig. 7d and 7e**). Also, we detected curtailed Mtb-specific CD4⁺ and CD8⁺ T cell function by multiple poly I:C injection and it is rescued by type I IFN receptor blockade (**Supplementary Fig. 7f**). In addition to Mtb-specific T cell responses, we also analyzed alteration of CXCL9 expression in myeloid cells by multiple poly I:C injection. Consistent with LCMV Arm infection, when mice were injected with poly I:C, expression of CXCL9 was reduced and it also reinvigorated by type I IFN receptor blockade in CD11b⁺F4/80⁺ cell populations (**Supplementary Fig. 13c and 13d**). Furthermore, we analyzed type I IFN levels in the serum and lung during LCMV infection or poly I:C injection. However, even if poly I:C was injected three times, the phenotype of severe pulmonary lesions observed in the coinfection group was not exhibited because the amount of type I IFN produced by LCMV infection was significantly higher than that produced by poly I:C injection (**Supplementary Fig. 8a and 8b**). Therefore, we suggested that although the poly I:C injection model cannot phenocopy the LCMV coinfection effect due to the limitation of the amount of type I IFN produced by poly I:C injection, lung immunopathology during Mtb infection was consequently dependent on the type I IFN amount *in vivo*. We modified the text in the revised manuscript as follows.

“To induce enhanced and sustained type I IFN signaling, we intratracheally injected poly I:C into mice three times at 14, 15 and 16 days post Mtb infection (Supplementary Fig. 7a). At 26 days post Mtb infection, the pulmonary lesions were deteriorated after multiple poly I:C treatments, and this effect was dramatically alleviated by type I IFN receptor blockade (Supplementary Fig. 7b and c). In addition, the reduction in body weight in the multiple poly I:C-injected group was severe, which was not observed when the type I IFN receptor was

blocked (Supplementary Fig. 7d). These results were also supported by changes in the bacterial loads among three groups (Supplementary Fig. 7e). In aspects of Mtb-specific T cell responses, IFN- γ -producing Mtb-specific CD4⁺ and CD8⁺ T cells were curtailed by poly I:C injection, but these reduced functionalities were reinvigorated by type I IFN receptor blockade (Supplementary Fig. 7f). Multiple poly I:C injections exacerbated pulmonary pathology, but the extent was less than that with viral coinfection. Additionally, the IFN- α levels and their sustainability in serum and lung lysates were significantly higher in LCMV Arm-infected mice than in triple poly I:C-injected mice (Supplementary Fig. 8a and 8b), which was correlated with the degree of pulmonary lesion exacerbation. Hence, the degree of pathology exacerbation was dependent on type I IFN levels and sustainability in vivo.” [Lines 283-298]

“Furthermore, when mice were injected with multiple poly I:C, they exhibited the reduced expression of CXCL9 in CD11b⁺F4/80⁺ populations and this curtailed expression of CXCL9 was rescued by type I IFN receptor blockade (Supplementary Fig 13c and 13d), which suggest that type I IFN induced by poly I:C injection inhibited the expression of CXCL9 in vivo.” [Lines 475-479]

3. Figure 4 e-f. Although the heatmap in 4e does appear to show that Cxcl9 and Cxcl10 expression are decreased in the macrophage 2 cluster during coinfection, and that expression of these chemokines can partly be rescued by α -IFNAR treatment, the violin plots in 4f do not look all that convincing. Is there another way to display this data?

Response: We apologize for our insufficient display of our previous violin plots showing CXCL9 and CXCL10 expression. We modified the plots to clearly show the difference in chemokine expression among groups in the Macrophage-2 cluster (**Fig. 4e**). Additionally, we added individual dots to the modified violin plots to more convincingly present the data.

4. Further, the relevance of this macrophage cluster is not quite clear. Is this subset the predominant source of Cxcl9 and Cxcl10 during Mtb infection? Would it be helpful to compare Cxcl9 and Cxcl10 expression among the macrophage, monocyte, and neutrophil clusters during Mtb infection alone? Is it possible to assess whether the macrophage 2 cluster has increased

IFN- γ -STAT1 signaling compared to the other myeloid clusters, which may contribute to its high expression of Cxcl9?

Response: We appreciate the reviewer's suggestion to clarify the predominant sources of CXCL9 and CXCL10. As the reviewer suggested, we compared Cxcl9 and Cxcl10 gene expression among the macrophage, monocyte, and neutrophil clusters during Mtb K infection alone (**Supplementary Fig. 10a**). As a result, we found that the Macrophage-2 cluster was the predominant source of Cxcl9 and one of the dominant sources of Cxcl10, along with the Neutrophil-2 cluster. In addition to analysis of scRNA-seq data, we further checked the predominant source of CXCL9 at the protein level by flow cytometry analysis. Among myeloid cell clusters, CXCL9 was predominantly expressed in the CD11b⁺F4/80⁺ cell population rather than the CD11b⁺Ly6G⁺ and CD11b⁺Ly6C⁺ cell populations (**Supplementary Fig. 10b and 10c**). We have added text to the revised manuscript as follows.

“Interestingly, the Cxcl9 and Cxcl10 genes were dominantly expressed after Mtb infection (Fig. 4d), and the Macrophage-2 (c3) cluster was the predominant source of Cxcl9 and one of the dominant sources of Cxcl10, along with the Neutrophil-2 cluster among myeloid clusters (Supplementary Fig. 10a).” [Lines 325-328]

“In addition to scRNA-seq analysis, we also investigated the predominant source of CXCL9 at the protein level in myeloid cell populations by flow cytometry analysis. In the Mtb K alone-infected group, CD11b⁺F4/80⁺ cells were the dominant cell type expressing CXCL9 (Supplementary Fig. 10b and 10c). We also found that the frequency and number of CD11b⁺F4/80⁺CXCL9⁺ cells significantly decreased in the coinfecting group but this decrease was rescued by type I IFN receptor blockade (Supplementary Fig. 10b and 10c).” [Lines 330-336]

5. Figure 4h. Do the effector CD4 T cell clusters express Cxcr3 and Cx3cr1? Is there a difference in expression of these Th1 molecules between experimental conditions?

Response: As the reviewer asked, we additionally analyzed the gene expression of Cxcr3 and Cx3cr1 in effector CD4 T cell clusters (**Supplementary Fig. 10d**). In particular, Cx3cr1 expression decreased in the coinfecting group and was partially increased by type I IFN receptor

blockade. However, Cxcr3 expression was not largely affected by type I IFN receptor blockade at the RNA level. We have added text to the revised manuscript as follows.

“Cx3cr1 expression in effector CD4 T cell clusters was downregulated in the coinfecting group and was partially rescued by type I IFN receptor blockade (Supplementary Fig. 10d)” [Lines 354-356]

6. Figure 6: The generation and use of P25 transgenic CD4⁺ T cells is a great approach to study how co-infection and type I IFN signaling impacts CD4 T cell differentiation, migration, and function. However, expression of CD44, ki67 and cytokines IFN- γ and TNF- α are the only parameters assessed here, and additional characterization of these Mtb-specific CD4 T cells would be helpful. Is there a difference in expression of CXCR3, CX3CR1, T-bet etc.? Given that a central component of the study is focused on CXCR3-ligands Cxcl9 and Cxcl10, expression of CXCR3 on P25 cells should be shown for all experimental groups, and a comparison of other Th1-associated markers would be helpful. If the authors do not have this data on hand, they should at least show the expression of these markers in their scRNA seq data for all experimental groups.

Response: We appreciate the reviewer’s comments. As the reviewer suggested, we newly analyzed the levels of the Cxcr3 and Cx3cr1 genes by scRNA-seq (**Supplementary Fig. 10d**). In addition to RNA level analysis, we also checked the protein expression levels of CXCR3 and T-bet by flow cytometry (**Supplementary Fig. 12b and 12c**). Unlike the pattern of Ki-67 and CD44 expression in P25 cells, there was no significant difference in T-bet expression in P25 cells. CXCR3 expression on P25 cells slightly decreased in the coinfecting group and tended to be rescued by type I IFN receptor blockade. We have added text to the revised manuscript as follows.

“Given our earlier data regarding type I IFN-dependent regulation of CXCL9 and CXCL10 (Fig. 4e and Fig. 5c), we checked the expression of their chemokine receptor CXCR3 and T-bet, a Th1-associated transcription factor. There was no significant difference between three groups, similar to the expression of Ki-67 and CD44 in P25 cells, although CXCR3 expression slightly decreased in the coinfecting group, which tended to be rescued by type I IFN receptor

blockade (Supplementary Fig. 12b).” [Lines 412-417]

7. Additionally, do the authors have data showing the kinetics of P25 cells in the lungs over time? Is there a major influx/accumulation of P25 cells in the lungs between 2-3 weeks post infection, which would fit with their overall model that coinfection with LCMV starting at 3 weeks post Mtb infection has no impact on pathology, likely due to establishment of robust Th1 responses by that time, whereas mice are still susceptible at 2 weeks post-infection which the authors suggest is due to insufficient Th1 responses at that time.

Response: We appreciate the reviewer’s valuable comments. As the reviewer recommended, we additionally checked the frequency and number of P25 cells at 2 and 3 weeks after Mtb infection (**Supplementary Fig. 12a**). We found that the frequency and number of P25 cells at 2 weeks post Mtb infection were much lower than those at 3 weeks post Mtb infection. As the reviewer commented, in our coinfection model, the mice coinfecting with LCMV at 2 weeks post Mtb infection exhibited severe immunopathology and a high bacterial burden due to insufficient Th1 responses at that time, whereas the mice coinfecting with LCMV at 3 weeks post Mtb infection showed no impact because of the establishment of robust Th1 responses by that time. We have added text to the revised manuscript as follows.

“In addition, we further analyzed the kinetics of P25 cells in the lungs. The frequency and number of P25 cells at 3 weeks post Mtb infection were higher than those at 2 weeks post Mtb infection, which suggested that a major influx and accumulation of P25 cells occurred between 2 and 3 weeks after Mtb infection (Supplementary Fig. 12a). This accumulation tendency would fit our overall model of coinfection with LCMV starting at 3 weeks post Mtb infection, showing no impact on pathology and indicating that establishment of robust Th1 responses could alleviate coinfection-mediated pulmonary pathology and bacterial outgrowth (Supplementary Fig. 3h-l).” [Lines 401-408]

8. As Cxcl10 is a known type I IFN-stimulated gene, it is somewhat surprising that Cxcl10 levels increase in the lungs after IFNAR blockade. Can the authors comment on this more in the discussion section? Is it thought that the increase in IFN- γ production upon IFNAR blockade is sufficient to amplify Cxcl10 production even in the absence of IFNAR signaling?

Response: Thank you for this valuable comment. We absolutely agree with the reviewer's opinion that prominent expression of CXCL10 upon type I IFN receptor blockade might be induced by abundant IFN- γ production in the absence of type I IFN signaling. Moreover, the type I IFN and IFN- γ signaling pathways are known to share phosphorylated-STAT1 transcription factor. Therefore, since type I IFN signaling was inhibited by type I IFN receptor blockade, most phosphorylated-STAT1 was used to strengthen the IFN- γ signaling pathway for CXCL10 expression. We have included the reviewer's comment and our response in the Discussion section of the revised manuscript as follows.

“Furthermore, although CXCL10 gene is well known as a type I IFN-stimulated gene, it was somewhat surprising that its expression was elevated in the lung after type I IFN signaling blockade in our coinfection model. However, given that type I IFN receptor blockade increased IFN- γ levels of T cells in the lung of coinfecting mice, it is possible that the effect of the increased IFN- γ production of T cells for CXCL10 production is greater than that of reduced type I IFN signaling, resulting in an increase in CXCL10 expression in the lung. In addition, the type I IFN and IFN- γ signaling pathways has been known to share the phosphorylated-STAT1 transcription factor. Therefore, since the type I IFN signaling was inhibited by type I IFN receptor blockade, most of phosphorylated-STAT1 would be used to the strengthen IFN- γ signaling pathway for CXCL10 expression.” [Lines 599-608]

Reviewer #3 (Remarks to the Author):

This report investigates Mycobacterium tuberculosis and acute lymphocytic choriomeningitis virus co-infection in mice. Mtb infection is a significant global public health concern and developing an understanding the impact of co-infections on the outcome and severity of disease is important for both deciphering the mechanisms of pathogenesis as well as for devising strategies to alleviate the infection. The authors tackle this by establishing Mtb infection in mice and subsequently eliciting an acute viral infection with LCMV. Co-infection with LCMV during a precise window of time, corresponding to the immune recruitment phase during Mtb, increases the severity of the Mtb infection when certain clinical isolates but not laboratory isolates are used. Further analyses of clinical Mtb isolate and acute LCMV co-infection demonstrates that this scenario results in increased neutrophilia and decreased Mtb-specific T cell responses. These alterations are somewhat reversible by blocking type-I IFN. The roles of type-I IFN in enhancing pathogenesis are further supported by experiments in which poly I:C was administered. One of the major changes that occurred during acute LCMV infection was to the inflammatory infiltrate and the expression of certain chemokines including CXCL9 and CXCL10, which impact the recruitment of Mtb specific T cells to the lung. Overall the studies highlight a potential role of type-I IFNs in enhancing Mtb pathogenesis by altering the inflammatory infiltrate, T cell response, and composition of the cytokine milieu.

Although the report contains a wealth of data and presents some interesting results, my level of enthusiasm for the report is mixed. Although model systems can be highly informative, I find this strategy odd. The disease enhancement phenomena is only observed with certain isolates of Mtb and it is not clear why at least some amount of augmentation is not observed with the non-clinical strains? It is interesting, but arguably under-developed, why the enhancement is so dependent on the timing of LCMV infection. Notwithstanding the aforementioned concerns, given the data presented it is also likely that factors other than type-I interferon levels also impact the outcome of Mtb infection.

Response: We deeply appreciate the reviewer's comments. We also find it interesting to note why the disease enhancement phenomenon is only observed with certain isolates of Mtb. As the reviewer indicated, this point is underdeveloped, and unfortunately, we believe that this aspect is beyond of the scope of this research in that we focused on the negative role of viral coinfection-induced sustained type I IFN production in delaying Mtb-specific Th1 responses

in the lung, resulting in disease enhancement. However, given that induction of IFN- β production by Mtb in macrophages is bacterial strain dependent via the cGAS/STING pathway, with unknown additional pathways that contribute to higher IFN- β induction (The Mechanism for Type I Interferon Induction by *Mycobacterium tuberculosis* is Bacterial Strain-Dependent; PLoS Pathog; 12(8):e1005809; 2016), this point still deserves further investigation. Although these attached data have not been reflected in the revised manuscript, we have conducted additional *in vitro* experiments to initiate further studies by employing four different Mtb strains: H37Ra, H37Rv, K, and M2. RAW264.7 cells were infected with these different Mtb strains at a multiplicity of infection (MOI) of 1. Interestingly, the K and M2 strains that mediate disease enhancement upon viral coinfection ranked in the top 2 among the employed Mtb strains by inducing higher expression of IFN- β by 7.4-fold and 14.7-fold, respectively, at 12 h post infection in RAW264.7 cells (a). When the RAW264.7 cells were treated with IFN- γ at 12 h post infection, the CXCL9 production significantly decreased in K- and M2-infected RAW264.7 cells compared to that in H37Ra- and H37Rv-infected RAW264.7 cells (b), indicating that IFN- γ -mediated CXCL9 expression correlates negatively with bacterial strain-dependent type I IFN signaling. We are currently investigating whether certain Mtb isolate-specific IFN- β induction accompanied by CXCL9 downregulation could mediate disease enhancement in synergy with viral coinfection-induced type I IFN production.

Additional points that the authors should consider are listed below:

1. The authors conclude that increased IFN-I levels alter the innate and adaptive response to Mtb. What remains unclear is whether IFN-I is acting directly or indirectly on the responses. For example, are T cell responses diminished due to IFN-I induced changes to myeloid cells or does the direct action of IFN-I on the priming T cells also impact the response etc?

Response: We carefully considered the reviewer's valuable comments. As the reviewer pointed out, it is important to determine whether type I IFN signaling acts directly or indirectly on the T cell response in our coinfection model. It is well known that both innate and adaptive immune cells ubiquitously express type I IFN receptors on their surface and that their function can be regulated via type I IFN signaling during infection with bacteria, viruses, or other pathogens (type I interferons in infectious disease; Nat. Rev. Immunol; 15:87-103; 2015). To validate the direct effect of type I IFN signaling on the T cell response in our coinfection model, we could generate spatiotemporal conditional knockout systems, such as $Ifnar1^{fl/fl} \times CD4^{Cre-ERT2}$ and $Ifnar1^{fl/fl} \times LysM^{Cre-ERT2}$, to knock out type I IFN receptors in CD4 T cells or myeloid cells at 2 weeks post Mtb infection by tamoxifen injection. However, this approach takes too much time and effort to technically generate these knockout mice and perform appropriate experiments. In our current study, it is conceivable that type I IFN signaling might downregulate IFN- γ -mediated expression of CXCL9 and CXCL10 in innate cells, thereby hindering the migration of Mtb-specific T cells into the lung and finally leading to severe TB pathogenesis. In addition to this possibility, T cells could also be directly altered by type I IFN signaling. In a previous study, type I IFN signaling directly influenced the clonal expansion of CD4⁺ T cells during LCMV infection. However, the direct effect of type I IFNs on Mtb-specific T cells during TB is still unclear, and thus, further studies using conditional knockout mice should be conducted to determine whether the direct action of type I IFN on Mtb-specific T cell responses will affect TB pathological exacerbation. We have added the following text to the Discussion section of the revised manuscript in consideration of the reviewer's valuable comments.

“However, in our coinfection model, it remains unclear whether type I IFN signaling is acting directly or indirectly on the T cell response. It is well known that both innate and adaptive immune cells are affected by type I IFN signaling during bacterial or viral infection because both cell types ubiquitously express type I IFN receptor in their surface⁵⁶. According to our current study, it is conceivable that type I IFN signaling might downregulate IFN- γ -mediated

expression of CXCL9 and CXCL10 in innate immune cells, thereby hindering the migration of Mtb-specific T cells into the lung and finally leading to severe TB pathogenesis. In addition to this possibility, T cells also could be directly altered by type I IFN signaling. In a previous study, type I IFN signaling had a direct influence on the clonal expansion of CD4⁺ T cells during LCMV infection⁵⁷. However, the direct effect of type I IFNs on Mtb-specific T cells during tuberculosis is still unclear, and thus, further studies should be conducted to determine whether the direct action of type I IFN on Mtb-specific T cell responses will affect TB pathological exacerbation.” [Lines 518-530]

2. As the authors acknowledge CXCL9 and CXCL10 levels were partially restored by IFNAR blockade (lines 317-320). This is an important point. Blocking type-I IFN in the co-infected animals clearly does not fully rescue the responses and revert them to those observed with Mtb alone. This is suggested by fig 4e, but also 4f, where although the anti-IFNAR treatment does change the response, it appears different to that elicited by Mtb only and the Mtb with LCMV coinfection. Similarly, in figure 5 the responses are also not fully rescued. While I agree that there is a likely temporal contribution of IFN-I there are very likely other factors involved. Thus, important gaps remain in our understanding of this phenomena which are not resolved by the current report.

Response: We deeply appreciate the reviewer’s comments. As the reviewer pointed out, Mtb-specific T cells are not fully rescued by type I IFN receptor blockade. We also agree that some type I IFN-independent mechanism stimulated by LCMV infection may also contribute to exacerbation of pulmonary pathology. As shown in **Supplementary Fig. 5**, we conducted several experiments by using diverse monoclonal antibodies to validate the effect of some immunological factors, which were enriched due to LCMV infection in our coinfection model. Unfortunately, no other factors contributing to exacerbation of pathology other than type I IFN were found, which suggests that pulmonary pathology in the coinfecting group was derived from the sum of many factors, and the main effector among them seems to be type I IFNs. Since the risk of TB-associated mortality due to coinfection with other infectious agents has been increasingly reported, further research should be conducted with diverse mouse coinfection models. We have included the reviewer’s concern as a limitation of our study in the Discussion section of the revised manuscript as follows.

“Furthermore, by using IFNAR-1 blockade and complementary application of type I IFN receptor knockout mice, we clearly show that much of the exacerbated pathology caused by coinfection is due to type I IFN signaling. However, Mtb-specific T cell responses or expression of CXCL9 and CXCL10 was not fully restored by IFNAR-1 blockade, which suggests that some type I IFN-independent mechanisms stimulated by LCMV infection may also contribute to exacerbated pulmonary pathology in coinfecting mice. Since the risk of TB-associated mortality due to coinfection with other infectious agents has been increasingly reported, further research should be conducted with diverse mouse coinfection models.” [Lines 531-538]

3. In figure 2E, IFN-gamma levels are analyzed at 21 days and are less than 2-fold different in the co-infection group. Is this difference sufficient to account for the alterations in bacterial clearance etc, and is this a transient or more substantial change? What is the profile if levels are checked over time?

Response: As the reviewer suggested, we newly measured IFN- γ levels in the lung over time during coinfection (**Supplementary Fig. 2e**). At 14 days post Mtb K infection (before LCMV Arm infection), IFN- γ was rarely detected in the lung. At 7 days post LCMV Arm infection, coinfecting mice had a lower level of IFN- γ in the lung than Mtb K-infected mice. At 14 days post LCMV Arm infection, the total IFN- γ level in coinfecting mice was slightly higher than that in Mtb K alone-infected mice. In addition to total IFN- γ levels, we also analyzed ESAT6-specific T cell responses by restimulation with an ESAT6 peptide pool (**Supplementary Fig. 2f**). Consistent with the total IFN- γ levels, comparison of the Mtb-specific T cell responses between the Mtb K alone-infected group and the coinfecting group revealed similar functionality at 14 days post LCMV Arm infection, suggesting that the ESAT6-specific T cell response was delayed at 7 days post LCMV Arm infection. Therefore, our data emphasized the importance of the timely presence of Mtb-specific IFN- γ -producing Th1 cells in Mtb-infected lung tissues to prevent viral coinfection- and/or type I IFN-induced TB immunopathology. We have added text to the revised manuscript as follows.

“In addition, we further analyzed the kinetics of IFN-g and ESAT6-specific T cell responses. At 14 days post LCMV Arm infection, both total IFN-g and ESAT6-specific T cell responses in the lung were similar between the Mtb K alone-infected and coinfecting groups (Supplementary

Fig. 2e and 2f), suggesting that ESAT6-specific T cell responses were transiently delayed at 7 days post LCMV Arm infection. Taken together, these data suggest that the absence of Mtb-specific IFN- γ -producing T cells and the increased neutrophil levels contribute to the severity of TB outcomes in Mtb-virus coinfecting mice.” [Lines 165-171]

4. In figure 2F, the source of the specimens shown in the upper and lower panels should be clearly indicated.

Response: We apologize for our insufficient description. We have added the source of the specimen [**Mtb K (upper panels), Mtb K→LCMV (lower panels)**] in **Fig. 2f**.

REVIEWERS' COMMENTS

Reviewer #1 (Remarks to the Author):

The authors addressed my concerns

Reviewer #2 (Remarks to the Author):

The authors have done an excellent job in revising the manuscript and addressing the major concerns that I had. I appreciate that the authors performed several additional experiments and analyses that in the end really come together to support the overall conclusions of the manuscript. The new Poly I:C experimental data shown in Supplemental Figure 7, really do help support the model that robust and sustained type I IFN induction during the second week of Mtb K infection promotes pulmonary pathology and that this can be rescued upon IFNAR blockade. Additionally, mapping out the kinetics of P25 cells in the lungs also showed a robust expansion of Mtb-specific CD4 T cells between 2 to 3 weeks post infection, which further supports the authors conclusions that establishment of robust Th1 responses by 3 weeks post Mtb infection likely explains why co-infection with LCMV at this later time-point does not result in pathology. I do not have any remaining concerns and believe that this study will be an important contribution to the field.

Reviewer #3 (Remarks to the Author):

This is well written resubmission which shows that an early acute viral co infection results in diminished Mtb specific CD4 responses and increased neutrophil levels, and exacerbates Mtb pathogenesis. Attributed to increased type I interferon and reduced CXCL9 levels. My main concerns remain with the physiological relevance of the experimental system and results. Although LCMV is a natural mouse pathogen, Mtb is not and it is unclear how this coinfection model will reflect natural Mtb infection in humans. Notwithstanding the large data set I do remain skeptical about how meaningful the findings are. In addition to my overall concern about the experimental approach there are a few other points that are not fully resolved, as outlined below.

1. As the authors acknowledge factors other than type I interferon appear to exacerbate the responses, but these remain unresolved (but possible very important?).
2. It also remains unresolved whether the interferon response is acting directly or indirectly (or both) to regulate the T cell response.
3. It is laudable that the authors evaluated multiple Mtb strain. Nevertheless, it remains unclear why the exacerbation of disease phenomena appears dependent on the Mtb strain (Figure 1h-j). This brings into question the general applicability of the findings, possibly undermining the value of the model system?

Although there are unresolved issues, at this stage I suspect that it is time to move on and have these findings reported.

Responses to reviewer's comments

Reviewer #1 (Remarks to the Author):

The authors addressed my concerns

Response: We deeply appreciate the reviewer's response.

Reviewer #2 (Remarks to the Author):

The authors have done an excellent job in revising the manuscript and addressing the major concerns that I had. I appreciate that the authors performed several additional experiments and analyses that in the end really come together to support the overall conclusions of the manuscript. The new Poly I:C experimental data shown in Supplemental Figure 7, really do help support the model that robust and sustained type I IFN induction during the second week of Mtb K infection promotes pulmonary pathology and that this can be rescued upon IFNAR blockade. Additionally, mapping out the kinetics of P25 cells in the lungs also showed a robust expansion of Mtb-specific CD4 T cells between 2 to 3 weeks post infection, which further supports the authors conclusions that establishment of robust Th1 responses by 3 weeks post Mtb infection likely explains why co-infection with LCMV at this later time-point does not result in pathology. I do not have any remaining concerns and believe that this study will be an important contribution to the field.

Response: We appreciate that the reviewer recognizes the hypothesis-generating nature of our work. We also thank the reviewer for dedicating your valuable time to provide expert comments to improve the scientific quality of our manuscript.

Reviewer #3 (Remarks to the Author):

This is well written resubmission which shows that an early acute viral co infection results in diminished Mtb specific CD4 responses and increased neutrophil levels, and exacerbates Mtb pathogenesis. Attributed to increased type I interferon and reduced CXCL9 levels. My main concerns remain with the physiological relevance of the experimental system and results. Although LCMV is a natural mouse pathogen, Mtb is not and it is unclear how this coinfection model will reflect natural Mtb infection in humans. Notwithstanding the large data set I do remain skeptical about how meaningful the findings are. In addition to my overall concern about the experimental approach there are a few other points that are not fully resolved, as outlined below.

Response: We thank the reviewer for dedicating your valuable time to provide expert comments on our manuscript. According to the reviewer's comments and suggestions, we carefully reviewed the employed methods, the experimental approaches and scope of the study to provide additional explanations in the revised manuscript. Regarding the physiological relevance of the experimental system and results, it is true that Mtb is not a murine pathogen. Nevertheless, many studies for coinfection models of Mtb have been conducted in mouse model systems to figure out the immune mechanisms ^{1, 2, 3, 4, 5}.

In addition, the importance of type I IFNs on the pathogenesis of Mtb infection has been clinically and experimentally emerging in terms of TB susceptibility and progression ^{2, 6, 7, 8, 9, 10, 11, 12}. However, most of experimental studies have emphasized the role of type I IFNs during Mtb infection in view of innate immunity or its relevant cytokines. In contrast, our study focused on the impact of viral infection-induced IFNs on TB immunopathogenesis and revealed that impediment of T cell influx into in Mtb-infected lungs are a main cause of the progression to severe TB pathogenesis by the viral coinfection.

Not only that, but there are many clinical reports on coinfection of Mtb and viruses including SARS-CoV-2, HIV, and influenzae virus [**Lines 50-64**]. Although Mtb and LCMV coinfection cannot occur in humans, we attempted to reveal the underlying mechanisms on how and why viral coinfection exacerbate the pathogenesis of Mtb. We found that type I IFN signaling induced by virus coinfection during Mtb infection delayed the Mtb-specific Th1 cell response in the lung, thereby resulted in exacerbation of pulmonary pathology with uncontrolled Mtb

growth. Thus, we believe that our study provides a novel underlying mechanism in terms of T cell response for the aggravated pathogenesis of Mtb by viral coinfection and type I IFN signaling rather than innate immune responses.

1. As the authors acknowledge factors other than type I interferon appear to exacerbate the responses, but these remain unresolved (but possible very important?).

Response: We agree with the reviewer's opinion on factors other than type I IFNs to exacerbate the responses. However, according to antibody blocking and type I IFN receptor KO mouse experiments clearly demonstrated abrogation of immunopathology by LCMV coinfection during Mtb infection. Thus, we believe that exacerbated Mtb immunopathology by virus coinfection was principally evoked in a sustainability and degree of type I IFN-dependent manner, rather than another effect of virus. We will further investigate the roles of other factors in exacerbated responses of Mtb-LCMV coinfection.

2. It also remains unresolved whether the interferon response is acting directly or indirectly (or both) to regulate the T cell response.

Response: We appreciate the reviewer's comments. As the reviewer pointed out, it is important to resolve whether type I IFN signaling acts directly or indirectly on the T cell response in our coinfection model. In fact, it is difficult to figure out the direct or indirect effect of type I IFNs on T cells due to the lack of an appropriate *in vivo* experimental system that can regulate type I IFN signaling spatiotemporally. We have discussed the reviewer's point and study limitation by citing references in the revised manuscript.

“Furthermore, another report suggested that effects of type I IFN to CD8⁺ T cells may depend on the timing of type I IFN exposure. For example, if CD8⁺ T cells are exposed to type I IFN signaling before antigen encounter, type I IFNs would be harmful to T cell proliferation. However, if they are exposed simultaneously to antigen and type I IFNs, type I IFN signaling positively acted on T cell proliferation¹³. Since the direct or indirect effect of type I IFNs on Mtb-specific T cells during TB is still unclear, thus, further studies should be conducted to

determine whether the direct action of type I IFN on Mtb-specific T cell responses will affect pathological exacerbation of TB.” [Lines 545-552]

3. It is laudable that the authors evaluated multiple Mtb strain. Nevertheless, it remains unclear why the exacerbation of disease phenomena appears dependent on the Mtb strain (Figure 1h-j). This brings into question the general applicability of the findings, possibly undermining the value of the model system?

Response: We appreciate the reviewer’s comment. We also appreciate for the concern about Mtb strain-dependent exacerbated disease phenotype. As we discussed in the previous response to the reviewer, the levels of the preexisting type I IFNs or IFN- γ (Th1 response) may be the determinants for viral coinfection-induced immunopathology during Mtb infection since BCG immunization prevents this phenomenon. In addition, innate lymphoid cells producing IFN- γ such as ILC1 and NK(T) cells may be involved in the protection from this event. Furthermore, differential virulence (factors), genotypes, and genome contents of Mtb strains may be another factor determining this phenomenon. We recognize it is scientifically very important. Nevertheless, it needs another huge work to reveal the underlying mechanism. We believe that this aspect is beyond of the scope of the current our research. We will further investigate how and why Mtb-dependent immunopathology occur by viral coinfection based on the reviewer’s excellent comments.

4. Although there are unresolved issues, at this stage I suspect that it is time to move on and have these findings reported.

Response: We truly appreciate for your generous response on our study.

1. Artola-Borán, M. *et al.* Mycobacterial infection aggravates Helicobacter pylori-induced gastric preneoplastic pathology by redirection of de novo induced Treg cells. *Cell Reports* **38**, 110359 (2022).
2. Xu, W. *et al.* Early innate and adaptive immune perturbations determine long-term severity of chronic virus and Mycobacterium tuberculosis coinfection. *Immunity* **54**, 526-541. e527 (2021).
3. Ring, S. *et al.* Blocking IL-10 receptor signaling ameliorates Mycobacterium tuberculosis infection during influenza-induced exacerbation. *JCI insight* **4** (2019).
4. Monin, L. *et al.* Helminth-induced arginase-1 exacerbates lung inflammation and disease severity in tuberculosis. *The Journal of clinical investigation* **125**, 4699-4713 (2015).
5. Redford, P.S. *et al.* Influenza A virus impairs control of Mycobacterium tuberculosis coinfection through a type I interferon Receptor-Dependent pathway. *The Journal of infectious diseases* **209**, 270-274 (2014).
6. Firszt, R. & Vickery, B. An interferon-inducible neutrophil-driven blood transcriptional signature in human tuberculosis. *Pediatrics* **128**, S145-S146 (2011).
7. Zak, D.E. *et al.* A blood RNA signature for tuberculosis disease risk: a prospective cohort study. *The Lancet* **387**, 2312-2322 (2016).
8. Bénard, A. *et al.* B cells producing type I IFN modulate macrophage polarization in tuberculosis. *American journal of respiratory and critical care medicine* **197**, 801-813 (2018).
9. Tang, X. *et al.* Sustained IFN-I stimulation impairs MAIT cell responses to bacteria by inducing IL-10 during chronic HIV-1 infection. *Science advances* **6**, eaaz0374 (2020).
10. Olson, G.S. *et al.* Type I interferon decreases macrophage energy metabolism during mycobacterial infection. *Cell reports* **35**, 109195 (2021).
11. Barber, K.D.M. *et al.* Host-directed therapy of tuberculosis based on interleukin-1 and type I

interferon crosstalk. (2014).

12. Moreira-Teixeira, L. *et al.* Type I IFN exacerbates disease in tuberculosis-susceptible mice by inducing neutrophil-mediated lung inflammation and NETosis. *Nature communications* **11**, 1-18 (2020).
13. Marshall, H.D., Urban, S.L. & Welsh, R.M. Virus-induced transient immune suppression and the inhibition of T cell proliferation by type I interferon. *Journal of virology* **85**, 5929-5939 (2011).